# Atomic-scale phase separation induced clustering of solute atoms

Lianfeng Zou[1,9], Penghui Cao[2,9], Yinkai Lei[3,9], Dmitri Zakharov[4], Xianhu Sun[1], Stephen D. House[5,6], Langli Luo[1], Jonathan Li[1], Yang Yang[7], Qiyue Yin[1], Xiaobo Chen[1], Chaoran Li[1], Hailang Qin[1], Eric A. Stach[8], Judith C. Yang[5,6], Guofeng Wang[3] & Guangwen Zhou[1✉]

Dealloying typically occurs via the chemical dissolution of an alloy component through a corrosion process. In contrast, here we report an atomic-scale nonchemical dealloying process that results in the clustering of solute atoms. We show that the disparity in the adatom–substrate exchange barriers separate Cu adatoms from a Cu–Au mixture, leaving behind a fluid phase enriched with Au adatoms that subsequently aggregate into supported clusters. Using dynamic, atomic-scale electron microscopy observations and theoretical modeling, we delineate the atomic-scale mechanisms associated with the nucleation, rotation and amorphization–crystallization oscillations of the Au clusters. We expect broader applicability of the results because the phase separation process is dictated by the inherent asymmetric adatom-substrate exchange barriers for separating dissimilar atoms in multi-component materials.

[1] Department of Mechanical Engineering & Materials Science and Engineering Program, State University of New York, Binghamton, NY 13902, USA. [2] Department of Mechanical and Aerospace Engineering, University of California, Irvine, CA 92697, USA. [3] Department of Mechanical Engineering and Materials Science, University of Pittsburgh, Pittsburgh, PA 15261, USA. [4] Center for Functional Nanomaterials, Brookhaven National Laboratory, Upton, NY 11973, USA. [5] Department of Chemical and Petroleum Engineering, University of Pittsburgh, Pittsburgh, PA 15261, USA. [6] Environmental TEM Catalysis Consortium (ECC), University of Pittsburgh, Pittsburgh, PA 15261, USA. [7] Department of Nuclear Science and Engineering, Massachusetts Institute of Technology, Cambridge, MA 02139, USA. [8] Department of Materials Science and Engineering, University of Pennsylvania, Philadelphia, PA 19104, USA. [9]These authors contributed equally: Lianfeng Zou, Penghui Cao, Yinkai Lei. ✉email: gzhou@binghamton.edu

Tuning the composition of multicomponent materials, either through alloying or dealloying, offers an effective means to manipulate material properties. Often, alloying is a spontaneous process that depends on the intrinsic miscibility of the alloy components, as given by the well-known Hume-Rothery Rules[1–3]. In contrast, the reverse process, separating two miscible elements, is thermodynamically unfavorable because of the lack of a driving force for one component to leave the solid solution. Therefore, dealloying usually relies on chemical leaching to remove the less noble metal in a corrosive environment, leaving behind an altered residual structure. Today, the chemical dealloying approach has become very popular to synthesize various nanoporous materials using different metal alloy precursors including Cu–Au[4,5], Au–Ag[6–9], Au–Al[10,11], Au–Pd–Ag[12,13], Pt–Cu[14–17], Pt–Al[18–20], Pt–Fe[21,22], Au–Zn[23], and Cu–Mn[24–26]. The Cu–Au system fulfills the Hume-Rothery Rules and forms a face-centered cubic (FCC) solid solution through the full range of composition and temperature[27,28]. Therefore, the Cu–Au alloy is an ideal model system to investigate the phase separation behavior of the miscible systems. The Cu–Au system develops several intermetallic compounds that are stable to temperatures of ~ 390 °C. Our previous in situ TEM observations have shown that annealing Cu-10at.%Au solid solutions below the order–disorder transition temperature results in the formation of an ordered Cu₃Au-like surface alloy[29]. The resulting ordered surface alloy is found to have a profound effect on surface properties including acting as an effective barrier to inhibit dislocation annihilation at free surfaces[30] and slow down surface oxidation[31].

Herein, we report the self-demixing in the Cu–Au system via an atomic-scale phase separation process that differs completely from the chemical dealloying mechanism and thus represents a significant departure from the Hume-Rothery rules. Through the use of real-time transmission electron microscopy (TEM) and atomistic simulations, we show that Au atoms are kinetically separated from the Cu–Au solid solution at the surface and aggregate into Au clusters that exhibit a rich variety of dynamics resulting from the cluster and support interfacial interactions. The observed phenomena reported here are of considerable practical importance as the phase separation process has wide relevance for a broad range of material systems, properties, and reactions, which include metallurgy, nanostructure synthesis, and heterogeneous catalysis.

## Results

### Clustering of Au atoms in Cu–Au solid solution.
Cu-10at.%Au (100) single-crystal films with ~50 nm thickness are grown on NaCl(100) by e-beam co-evaporation (Supplementary Fig. 1). The as-prepared Cu–Au films are subsequently removed from the substrate by dissolution of NaCl in deionized water, washed, and mounted on a TEM specimen holder. The films have good continuity over large areas with uniform distribution of Cu and Au (Supplementary Fig. 2). Quantitative analysis of the EDS spectra obtained from the as-prepared Cu–Au film showed an atomic ratio of 90.7% Cu and 9.3% Au, which is close to the targeted composition by controlling the evaporation rate of the two electron guns. The Cu–Au films are then annealed at 350 °C and 1 × 10⁻³ Torr of H₂ gas flow to remove native oxides and generate faceted holes (Supplementary Fig. 3a), where the complete removal of native oxide is confirmed by electron diffraction (Supplementary Fig. 3b). Electron energy loss spectroscopy (EELS) experiments are also performed to ensure that the thin films annealed in H₂ gas flow are free of oxygen (Supplementary Fig. 3c, d).

The edges of faceted holes permit in situ TEM imaging of the phase separation processes in cross-sectional views. Upon annealing at 600 °C, the Cu–Au solid solution experiences an unexpected phase separation that results in the massive formation of nanoclusters on the planar surfaces and along the edges of the holes, as shown in Fig. 1a. Some nanoclusters on the planar surface regions show moire fringe contrast, indicative of misalignment or different lattice spacings between the particles and the support. The inset in Fig. 1a is a magnified view of a cluster formed along a hole edge, which shows an amorphous-like inner part in contact with the substrate and well-developed crystalline lattice planes for the outer part of the particle. Figure 1b illustrates a high-resolution TEM (HRTEM) image of a fully crystalized cluster on a (100) facet, showing distinctive lattice spacings from the parent phase, as evidenced by the presence of misfit dislocations. As viewed along the [001] zone axis, the cluster shows an FCC lattice symmetry and an interplanar spacing of 2.02 Å, which match the crystal lattice of Au and suggest a nonchemical dealloying process that results in the phase separation (Supplementary Note 2). The Au cluster on the (100) facet shows a pronounced three-dimensional (3D) island shape (Fig. 1b) whereas the cluster on the (110) facet tends to spread over the surface and adopts a 2D-like wetting-layer morphology (Fig. 1c). The formation of Au clusters is further confirmed by high-angle annular dark-field scanning transmission electron microscopy (HAADF-STEM) imaging and energy dispersive X-ray spectroscopy (EDS) composition analysis. As marked by the dashed lines in the low-magnification HAADF-STEM image (Fig. 1d), clusters are visible along the edge of a void, consistent with the E-TEM observations (Fig. 1a). The enlarged view is an atomically resolved HAADF-STEM image obtained from the cluster marked by the blue rectangle box in Fig. 1d, in which the measured lattice spacings match well with the interplanar spacings of Au(111) and (110). Meanwhile, the EDS linescan across the cluster as marked by the dashed arrow in Fig. 1d indicates that the cluster consists of 88% of Au and 12% Cu (atomic percentage). The measured high Au content confirms that the segregated clusters are dominated by Au atoms, consistent with the measured crystal lattice spacings in the HRTEM images (Supplementary Note 1 and Supplementary Fig. 4) and the HAADF image (inset in Fig. 1d) as well as the prediction from our KMC simulations as shown below in Fig. 2.

### Phase separation by adatom–substrate exchanges.
The observed phase separation is unexpected because Cu and Au atoms have a strong tendency to intermix, as predicted by the Hume-Rothery Rules. Meanwhile, the massive formation of Au clusters as shown in Fig. 1a does not fall into any predictions of current theoretical models of surface segregation[32–34]. This is because Cu and Au have strong tendency to form Cu–Au bonds and the pure surface segregation results in only a ~40% Au concentration in the topmost layer (Supplementary Fig. 5), which agrees with the reported maximum of ~50% of Au surface concentration for single-crystal Cu–Au alloys even with a higher bulk Au concentration of 25%[35–37]. Here we attribute this phase separation process to a kinetic mechanism that self-filtrates Cu atoms out of a fluid phase consisting of Cu and Au adatoms, resulting in significant enrichment of Au adatoms that subsequently aggregate into clusters. At the elevated temperature of 600 °C, the surface has many active sources including atomic steps, ledges, and kinks for the massive formation of Cu and Au adatoms via step-edge detachment[15,38–40]. This is experimentally confirmed by our in situ TEM observations showing the fast retraction motion of atomic steps on the planar surface, thereby resulting in a significant flux of mobile Cu and Au adatoms on the substrate surface (Supplementary Note 2 and Supplementary Fig. 6). In addition, the thermally induced voids in the Cu–Au films can

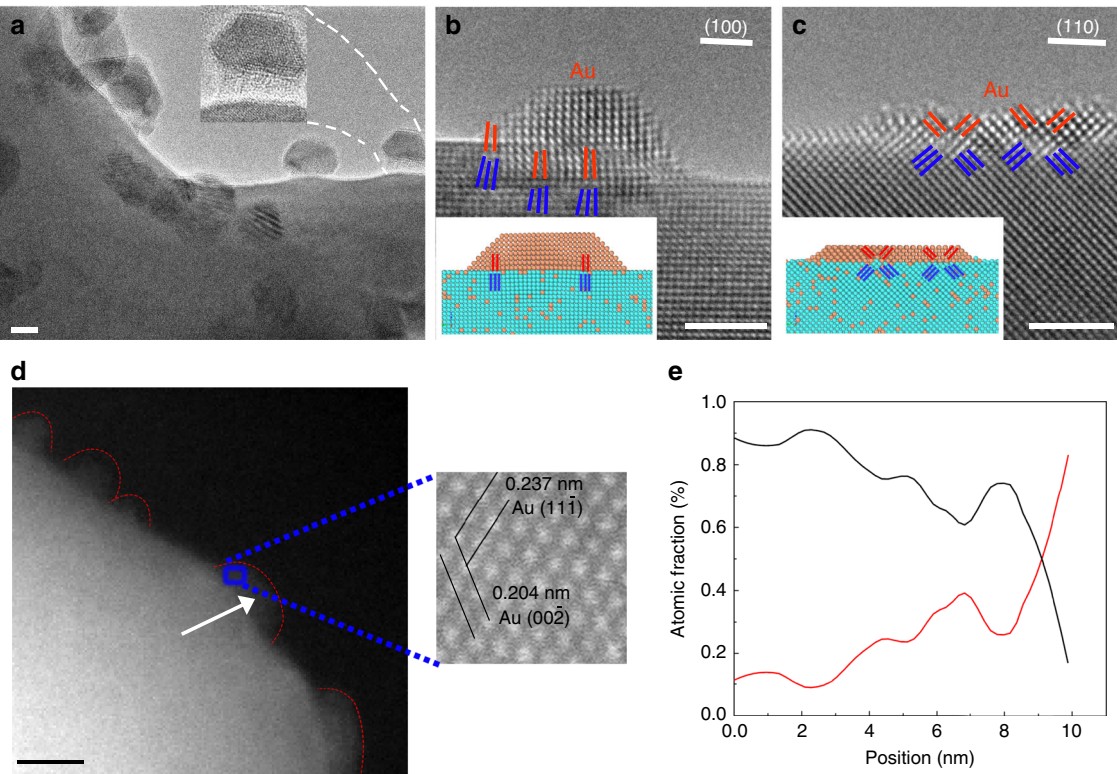

**Fig. 1 Phase separation and clustering of Au atoms in Cu–Au solid solution. a** TEM image of the Cu-10%Au thin film annealed at 600 °C and $1 \times 10^{-3}$ Torr of $H_2$ gas flow, the inset is a zoom-in view of a circled cluster. The inset is a zoom-in view of the cluster revealing that its inner part of the cluster is in an amorphous state. **b c** Equilibrium Au clusters on the (100) and (110) surfaces of the Cu(Au) solid solution. The lattice mismatch at the Au/Cu(Au) interface is marked by red and blue lines of the (200) atomic planes of the Au clusters and Cu(Au) substrate, respectively. Insets show schematically the lattice-mismatched Au clusters with the substrate. **d** Low-magnification HAADF-STEM image showing the formation of Au clusters (as marked by dashed lines) along a hole edge. The enlarged view is an atomically resolved HAADF-STEM image obtained from the cluster marked by the blue rectangle box. **e** STEM-EDS linescan along the arrow indicated in (**d**), confirming the high content of Au in the cluster. Scale bar, 5 nm (**a**, **d**), 2 nm (**b**, **c**).

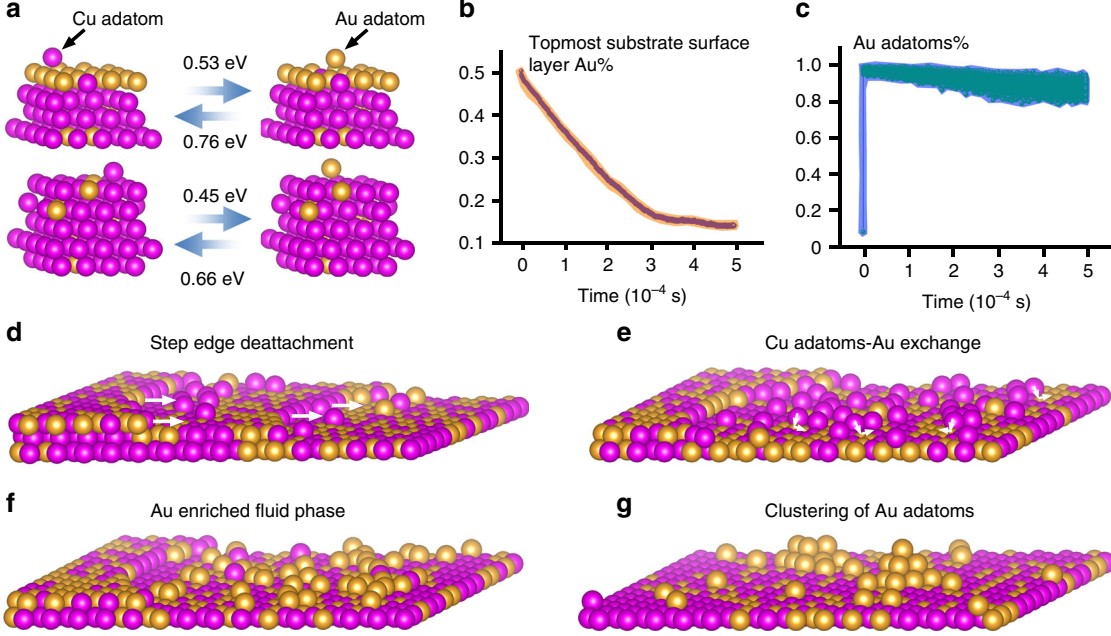

**Fig. 2 Adatom–substrate exchange induced phase separation in a Cu–Au fluid phase. a** NEB calculations of the adatom-substrate exchange barriers. Upper panel: exchange between a Cu adatom and an Au-rich substrate surface; lower panel: exchange between a Cu adatom and a Cu-rich substrate surface. **b**, **c** KMC simulations of the Au composition evolution in the topmost substrate surface layer and the fluid phase, respectively. **d–g** Schematic illustrating the phase separation process from the formation of a Cu–Au fluid phase of Cu and Au adatoms by step-edge detachments to the adatom–substrate exchange induced enrichment of Au atoms in the fluid phase and then to the clustering of Au adatoms (Supplementary Movie 1).

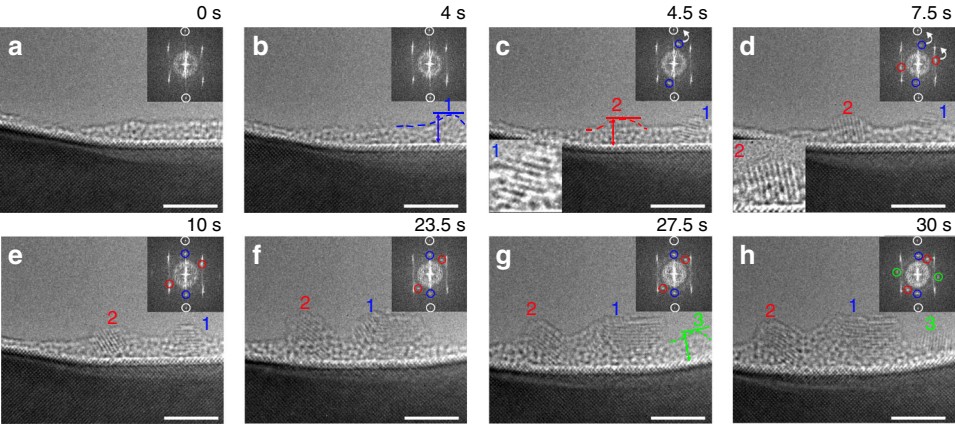

**Fig. 3 Amorphous-to-crystalline transition and grain rotation of Au clusters.** The in situ observations were performed at 600 °C and $1 \times 10^{-3}$ Torr of $H_2$ gas flow (Supplementary Movie 2). **a–d** The aggregation of Au atoms results in the formation of clusters 1 and 2 that subsequently transform into a crystalline state with the appearance of Au(111) lattice planes and cluster rotation. **e, f** Growth of crystalized clusters, where cluster 1 stays relatively stationary while cluster 2 undergoes slow rotation. The insets in (**c, d**) are the zoom-in view showing the presence of crystalline lattice in clusters 1 and 2. **g, h** Nucleation and growth of cluster 3 followed by subsequent amorphous-to-crystalline transition and grain rotation. Bottom-left: zoom-in view of clusters 1 and 2 in (**c, d**) respectively. Top-right: Fourier diffractograms, white rings circle out the (220) spot pair of the Cu(Au) solid solution; blue and red rings circle out the Au(111) diffraction spot pairs of clusters 1 and 2, respectively; and green rings circle out Au(200) diffraction spot pair associated with cluster 3. The white arrows in (**c, d**) mark the rotation direction of the clusters that results in the alignment of the (111) spot pairs of the Au clusters with the (200) spot pairs of the substrate. Scale bar, 4 nm (**a–h**).

grow larger due to the surface curvature effect on enhancing surface diffusion in the vicinity of the void edge[41]. Therefore, the edges of voids also act as an active source of forming mobile atoms to join the fluid phase of Cu and Au adatoms on the surface.

The observed formation of Au clusters is attributed to the asymmetric adatom–substrate exchange barriers that result in the enrichment of Au adatoms in the Cu–Au fluid phase. That is, the exchanges between Cu adatoms in the fluid phase and substrate Au atoms are more efficient than those between Au adatoms and substrate Cu atoms. As shown from our nudged elastic band (NEB) modeling, the energy barriers for the exchange of a Cu adatom in the fluid phase with the substrate Au atom in the Cu-rich and Au-rich surfaces are 0.45 and 0.53 eV, respectively, which are both smaller than the barriers (0.66 and 0.76 eV) for the exchanges of an Au adatom in the fluid phase with the substrate Cu atoms (Fig. 2a). This adatom–substrate exchange induced phase separation is further demonstrated by our kinetic Monte-Carlo (KMC) simulations by incorporating the disparity in the exchange barriers of Cu and Au adatoms in the fluid phase with the substrate atoms. Consistent with the predictions from the NEB results, Au atoms located in the substrate surface layer are gradually substituted by Cu adatoms in the fluid phase. We find that the Au concentration in the substrate surface drops from a starting composition of 50 to 14% after $4.9 \times 10^{-4}$ s (Fig. 2b). Meanwhile, the Cu–Au fluid phase evolves into a nearly pure Au fluid phase with an Au concentration fluctuating around 90% (Fig. 2c and Supplementary Fig. 7). The adatom–substrate exchange induced loss of Au atoms from the substrate surface can be continuously compensated by the surface segregation of Au atoms from the Cu–Au reservoir to the substrate surface for the sustained phase separation, driven by the lower surface energy of Au relative to Cu. Figure 2d–g illustrate schematically the overall process starting from the formation of a Cu–Au fluid phase by step-edge detachments to the adatom–substrate exchange induced enrichment of Au atoms in the fluid phase and then to the clustering of Au atoms in the fluid phase.

**Nucleation and rotation dynamics of Au clusters.** The resulting fluid phase of Au adatoms from this phase separation process serves as an ideal system to understand the clustering process of

adatoms. Figure 3 presents in situ HRTEM images of the nucleation of crystalline Au clusters, seen edge-on along the (110) facet of the substrate. A surface region with the intersection of two facets is selected for the in situ observations because the near corner regime may be the prominent site to trap adatoms (Fig. 3a). Initially, the aggregation of Au adatoms results in a small cluster that remains noncrystalline (i.e., cluster 1 marked with a dashed line in Fig. 3b). This is also evidenced by the diffractogram, in which only the diffraction spots associated with the Cu(Au) substrate are visible. However, when the cluster grows larger than a certain size (~2 nm), it transforms into a crystalline state (i.e., cluster 1 in Fig. 3c). This is shown by the presence of the crystalline lattice and the appearance of Au(111) diffraction spots (circled out by blue rings in the inset diffractogram), where the (111) lattice planes of the cluster are misaligned with the (110) surface of the substrate by ~12°. This amorphous-to-crystalline formation is further confirmed from a second cluster (cluster 2 in Fig. 3c) on the same surface. Likewise, cluster 2 first appears as an amorphous bump, but subsequently transforms into the crystalline state after growing thicker than ~2 nm, with the formation of crystalline lattice and Au(111) diffraction spots, as indicated by red rings in the inset diffractogram of Fig. 3d. Cluster 2 possesses a different in-plane orientation from cluster 1, as indicated by an ~78° misalignment of its (111) lattice planes with the (110) surface of the substrate. This suggests that as-crystalized clusters are randomly oriented. Upon crystallization, the small Au clusters undergo gradual rotation with the tendency to crystallographically align with the substrate. For instance, cluster 1 keeps rotating between 4.5 and 10 s (Fig. 3c–e), which results in the alignment of (111)Au//(220)Cu(Au). This can be confirmed by the inset diffractograms showing that the (220) diffraction spots of the Cu(Au) substrate circled out by the white rings become gradually aligned with the (111) spots of Au cluster (circled out by the blue rings). Similarly, cluster 2 shows the same trend to rotate towards the same alignment with the substrate (220) planes. This is evident from the diffractogram, in which the misalignment between the two (111) spots (marked with the red circles) of the cluster and the two (220) spots of the substrate (marked with the white circles) decreases from 78.5° to 38.4° at between 7.5 and 23.5 s (Supplementary Note 3 and

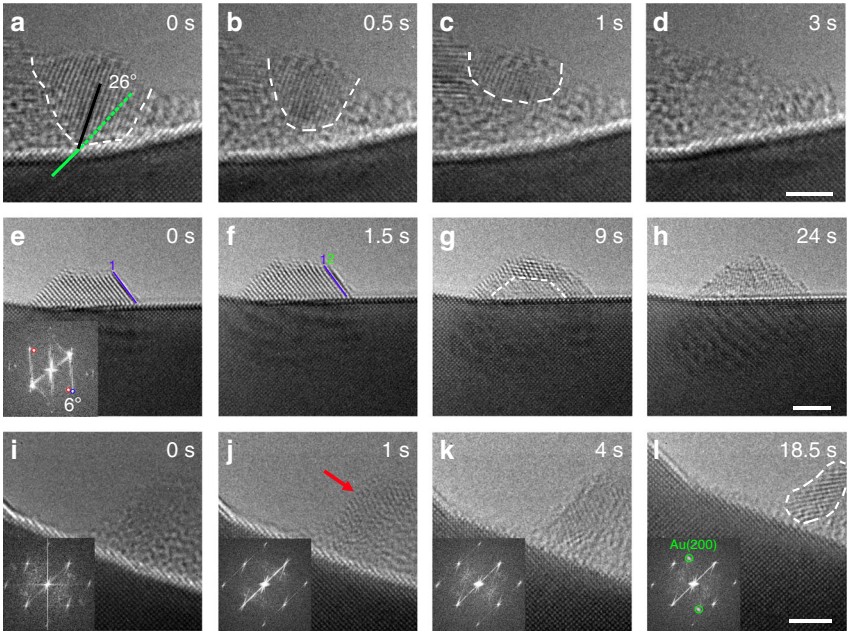

**Fig. 4 Amorphization–crystallization oscillations of supported Au clusters. a–d** In situ TEM image sequence showing the crystalline-to-amorphous transition for a significantly misaligned Au cluster. The black and green lines mark the misorientation angle between the Au cluster and the substrate. The white dashed lines outline the lower boundary of the crystalline Au cluster (Supplementary Movie 3). **e–h** In situ TEM image sequence showing the amorphization process in a slightly misaligned Au cluster (Supplementary Movie 4). Lines 1 and 2 mark the side surface of the Au cluster at 0 and 1.5 s, respectively. The white dashed lines outline the amorphized region of the Au cluster. Bottom-left inset in **e**: diffractogram showing the misalignment of the diffraction spots from the cluster (indicated by red and blue arrows). **i–l** In situ TEM images showing the recrystallization process in a large amorphous cluster (Supplementary Movie 5). The red arrow points to the surface area with partially ordered lattice fringes. The dashed white line circles out the crystallized area. Bottom-left insets are diffractograms, the diffraction spot pair marked by the green circles in (**l**) is associated with Au(200) lattice planes in the crystallized region of the cluster. All the in situ TEM images are captured in real time at 600 °C and $1 \times 10^{-3}$ Torr of $H_2$ gas flow. Scale bar, 2 nm (**a–l**).

Supplementary Fig. 8). However, the fast growth in size acts as an obstacle to further grain rotation and makes the larger cluster more resistant to rotation. Therefore, the rotation kinetics for cluster 2 gradually slow down after 23.5 s (Fig. 3f–h, and Supplementary Fig. 8), and the cluster fails to completely align with the substrate at the end of the sequence. By contrast, cluster 1 stays stationary without further rotation despite its smaller size (Fig. 3e–h). This indicates that the (111)Au//(220)Cu(Au) alignment is a preferred orientation between the cluster and substrate (the diffraction spots marked by blue and white circles in the diffractograms are constantly aligned after the alignment is established). Figure 3g, h shows the sequence of another cluster (labeled by 3) undergoing the similar amorphous-to-crystalline transition upon the cluster growth. Following the same transformation sequence as clusters 1 and 2, cluster 3 starts with the formation of an amorphous bump (labeled by 3) that subsequently transits into the crystalline state after reaching the lateral size of ~2 nm. All the three clusters show the similar size of ~2 nm to start the crystallization, which is consistent with the theoretically predicted critical size for the amorphous-to-crystalline transformation in Au, as shown in our simulations (Supplementary Note 4 and Supplementary Fig. 9). The same transformation pathway of all the three clusters shown here reveals the significant role of the cluster size in controlling the amorphous-to-crystalline phase transition as well as the cluster rotation kinetics.

**Amorphization–crystallization oscillations in Au clusters.** We refer the clusters that fail to crystallographically align with the substrate before growing more than a certain size as overgrown

clusters. These Au clusters are unstable and show further atom rearrangement. Figure 4a–d show a sequence of the structure evolution in an overgrown Au cluster. Initially, the crystalline cluster is misaligned with the substrate with an inclined angle of ~26°, as marked by the orientations of lattice planes between the cluster and substrate (Fig. 4a). This overgrown cluster (outlined by the dashed white lines in Fig. 4a), with a diameter of ~4 nm, stays anchored because of the insufficient driving force for the grain rotation to align with the substrate. Nevertheless, the cluster is observed to reach a more stable state through the crystalline-to-amorphous transition. The amorphization initiates from cluster–substrate interface sites (Fig. 4a, b), propagates toward the surface region (Fig. 4c), and eventually the entire cluster transforms into an amorphous state (Fig. 4d).

Figure 4e–h presents a sequence of HRTEM images showing another example of the structure evolution for a slightly misoriented cluster. The Au cluster, with a misorientation of ~6° relative to the substrate (the angle is determined by the measured clockwise rotation of the diffraction spots marked by blue and red rings in the inset diffractogram in Fig. 4e), displays the growth of new atomic planes at the beginning of the sequence, where a new single atomic layer (labeled 2 in Fig. 4f) is observed to grow on the existing facet (labeled 1 in Fig. 4e), further confirming the clustering of Au adatoms supplied from the fluid phase via surface diffusion. Because of its relatively large size, the overgrown Au cluster fails to align perfectly with the substrate and shows the amorphization starting from the cluster–substrate interface. After 9 s, while the majority of Au cluster maintains the good crystallinity, the crystalline region adjacent to the cluster–substrate interface becomes disordered with the presence of blurred lattice contrast, i.e., in the area

marked with the white dashed lines in Fig. 4g. Although the size and shape of the Au cluster remain relatively unchanged through time, the disordered lattice feature has propagated over the entire particle at the end of the sequence (Fig. 4h). However, the amorphous state does not represent a stable configuration for large clusters, a thermodynamic driving force exists for the amorphous-to-crystalline transition when the clusters are larger than the critical size, ~2 nm, as shown in Fig. 3. Figure 4i–l demonstrate such a recrystallization process in an amorphized cluster with a size larger than 5 nm. As shown in Fig. 4j, crystalline lattice planes start to appear from the upper surface region of the amorphous cluster (pointed by the red arrow in Fig. 4j), indicative of the preferred initiation of crystallization far away from the cluster–substrate interface region. This is opposite from the crystalline-to-amorphous transition that initiates from the cluster–substrate interface region (i.e., seen in Fig. 4a–d, e–h). As shown in Fig. 4k, l, the crystallized area expands toward the cluster–substrate interface and the crystalline lattice fringes become increasingly sharper with time, indicative of the improved crystallinity. This can be also reflected from the diffractograms, which show the presence of distinct and sharp Au (200) spots from the crystalized Au domain at 18.5 s (Fig. 4l), confirming the amorphous-to-crystalline transition in the areas outlined by the white dashed lines.

Completion of the amorphous-to-crystallization transition cycle can induce amorphization again if the crystalized cluster is still not well-aligned with the substrate. This has already been shown from the in situ TEM image sequences of two clusters in Fig. 4a–d, e–h. Therefore, there are two different mechanisms that lead to the crystallographically aligning Au clusters with the substrate: small clusters adjust their orientation via grain rotation (Fig. 3), whereas overgrown clusters tune their orientation via repeated crystallization–amorphization transformations to reach the stable orientations (Fig. 4). The crystallized Au clusters shown in Fig. 1b, c represent the stabilized orientations on the (100) and (110) surfaces after prolonged annealing, where the clusters are crystallographically well-aligned with the substrate and stabilized by the formation of misfit dislocation arrays. Figure 1b, c also shows that the equilibrium shape of clusters depends on the surface orientation of the substrate. The cluster on the (100) surface is stabilized as a 3D, dewetted shape, whereas the Au cluster on the (110) surface is stabilized as a 2D wetting-layer morphology. Accordingly, two distinct types of dislocation arrays are formed at the two different interfaces: the (110) Burgers vector for the (110) surface, whereas the (100) Burgers vector for the (100) surface, as shown by the colored lines in Fig. 1b, c. Analyses based on the Wulff construction reveal that the equilibrium shape of the crystallographically aligned Au clusters depends on two properties: inherent surface energies and interfacial dislocations (Supplementary Note 5 and Supplementary Fig. 10). On the one hand, the larger surface energy of the (100) surface than the (110) drives the clusters on the (100) surface to develop a 3D island shape that has a smaller total surface area than a 2D wetting-layer like morphology. On the other hand, the smaller Burgers vector (the [100]-type) for the (100) substrate releases less interface strain energy and thus favors the solid-solid dewetting, thereby resulting in an increased cluster height and 3D island morphology. By contrast, the larger Burgers vector (the [110]-type) releases more interfacial strain energy and thus makes the Au cluster to wet the (110) surface.

The amorphous–crystalline transitions and cluster rotation that we see are inherent and not significantly affected by the incident electron flux during the TEM observations. The possible electron beam effects including charging, heating, atom displacement, sputtering, and radiolysis[42], are concluded to be negligible in our observations (Supplementary Note 6), consistent with previous work[43]. Meanwhile, the amorphous state in Au clusters was also observed without electron beam irradiation, as evidenced in Fig. 1a, where a partially amorphized Au cluster was captured after the electron beam was blanked off. Similarly, the observed cluster rotation dynamics are induced by the cluster–substrate interfacial interactions and the electron beam effect is negligible. This is because the energy barrier for driving the rotation motion of a supported cluster is typically much higher than the energy barrier for inducing morphological changes of the cluster[44,45]. That is, if there is a significant electron beam effect, the cluster will undergo shape changes before (or simultaneously with) the cluster rotation. However, our in situ TEM observations show that Au clusters undergo rotation motions without obvious morphological changes, such as cluster 2 shown in Fig. 3e–h. This was further confirmed, as shown in Fig. 1b, c, by blanking off the electron beam, and the crystallographically aligned Au clusters were obtained after prolonged annealing without the electron beam irradiation. In addition, the $H_2$ gas flow has negligible influence on the observed phase segregation other than providing a reducing environment to maintain the surface cleanliness. This is confirmed by electron diffraction and EELS analyses showing the absence of oxygen in the Cu–Au film annealed in $H_2$ (Supplementary Fig. 3). This is also consistent with our ambient-pressure X-ray photoelectron spectroscopy (AP-XPS) measurements, which showed that any bulk-dissolved oxygen in Cu can be completely deoxygenated by the flow of $H_2$ gas at ~580 °C to form $H_2O$ molecules that spontaneously desorb from the surface, resulting in an atomically clean Cu surface at the elevated temperature[46]. To further confirm whether the $H_2$ gas has any effect on the surface segregation, we also employed AP-XPS to monitor the surface composition evolution of $Cu_3Au(100)$ during annealing under ultrahigh vacuum (UHV) and in 0.1 Torr of $H_2$ gas flow[47] (Supplementary Note 7). No noticeable differences in the surface composition can be observed between the UHV and $H_2$ gas flow[47] (Supplementary Fig. 11). This is consistent with other studies showing the high dissociation barriers of $H_2$ molecules on both Cu and Au surfaces[48–50]. Even for atomic hydrogen, it bonds weakly to Cu and Au, and desorbs from Au surfaces at the temperature above approximately −163 °C[51] and from Cu surfaces at the temperature of ~88 °C[52], both of which are much lower than the annealing temperature in our in situ experiments.

**MD simulations of structural oscillations in Au clusters**. To further substantiate the effects of the cluster–substrate interfacial interactions in driving the structure dynamics of the clusters, we perform molecular dynamics (MD) simulations to probe the structure stability of supported Au clusters under various interface misorientations. Three differently oriented Au clusters are constructed, including the 15°, 90°, and 0° misalignment of the (110) lattice planes of the cluster with the (110) surface of the substrate, respectively. Figure 5 captures the localized lattice evolution of the three clusters from the cross-sectional view. For the 15° misalignment (Fig. 5a), the cluster shows a structure instability by undergoing amorphization–recrystallization oscillations, consistent with the in situ TEM observations (Fig. 4e–h). The concurrent amorphization–recrystallization processes make the apparent amorphization relatively slow, however, the overall trend toward the amorphous state is evident, as indicated by the increased volume of the amorphous regions after 0.4 ns. The amorphous regions further propagate as the time elapses, concomitantly leading to the rotation of the remaining crystalline region (after 1 ns). Figure 5b illustrates the 90° misalignment case that results in the $(100)_{cluster}/(110)_{substrate}$ interfacial matching. Compared to the smaller misalignment (Fig. 5a), this cluster

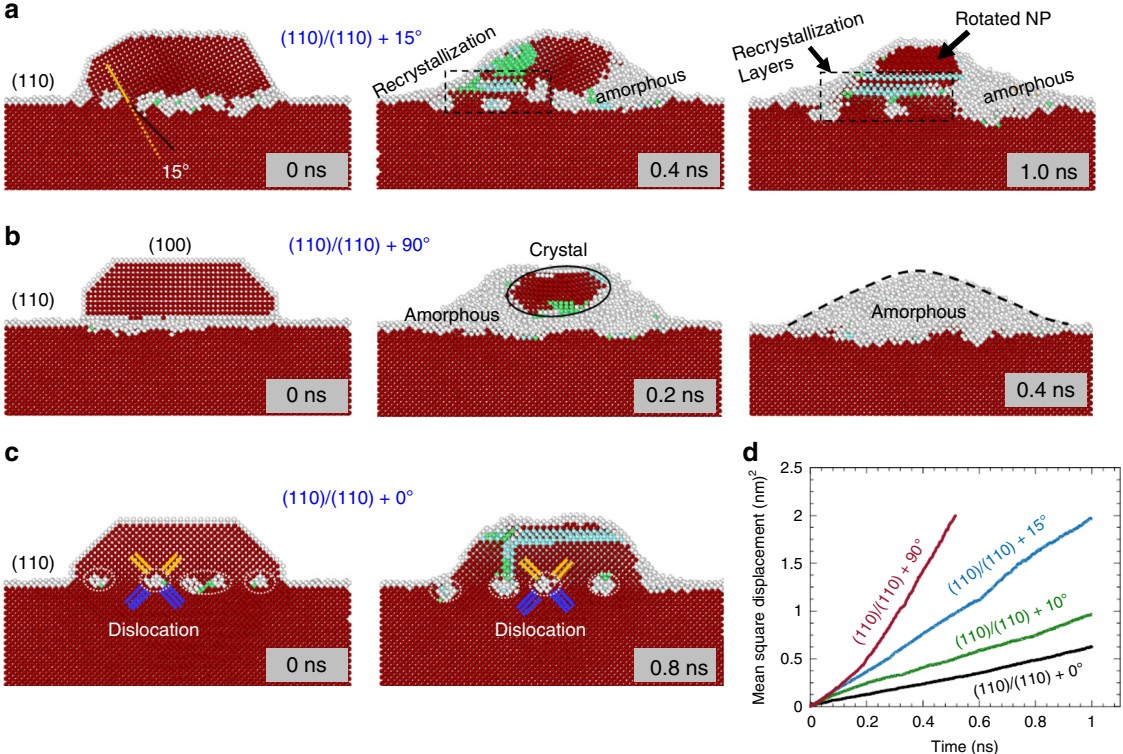

**Fig. 5 MD simulations of stability of supported Au clusters at 900 K. a–c** Atomic structure evolution in Au clusters that are misaligned with the substrate by 15°, 90°, and 0°, respectively, between the (110) lattice planes of the cluster and substrate (Supplementary Movies 6–8). **d** The MSD of atoms in the clusters measured as a function of time with the various interface orientations. The red, gray, green, and light blue spheres represent the atoms with FCC, disordered or unknown, BCC, and HCP structure.

undergoes accelerated amorphization. The whole cluster turns into the amorphous state at 0.4 ns, indicating a positive correlation between the misorientation angle and the amorphization kinetics. In contrast to the misaligned systems, Fig. 5c shows that the perfectly aligned cluster (i.e., $(110)_{cluster}/(110)_{substrate}$) is highly stable and the epitaxial cluster preserves its initial structure without experiencing noticeable amorphization. This is because the cluster is stabilized by the relaxation of the interfacial strain via the formation of misfit dislocations. This interfacial strain effect can also be evidenced by the misaligned cases (Fig. 5a, b), which show that the amorphization is initiated from the cluster–substrate interface region because of its higher strain whereas the recrystallization starts from the outer surface because of the weak influence from the interface. This interfacial strain effect on the structural dynamics of the clusters can be also evidenced by the measured statistic mean square displacement (MSD) of Au atoms in the clusters. As shown in Fig. 5d, the MSD increases more rapidly with the increase in the misalignment angle of the cluster with the substrate.

The in situ TEM observations in Figs. 3, 4 show that the dynamic structural evolution in the Au clusters has no noticeable effect on the underlying Cu substrate. That is, the interfacial strain induced structure oscillations (amorphalization and recrystallization) are limited to Au clusters. This is also confirmed from MD simulations (Fig. 5), which show that structure oscillations in the supported Au clusters do not induce noticeable structure changes in the Cu substrate. The higher stability of the substrate over supported Au clusters can be attributed to the size effect, where the small Au clusters are more susceptible to interfacial strain fluctuations because of their large surface/interface area-to-volume ratio and lower barriers for the rearrangement of atoms compared to the rigid substrate. This is

also consistent with the large body of literature demonstrating that epitaxial islands typically undergo a series of shape transitions induced by interfacial strains whereas the substrate remains relatively unaffected[53–56]. Instead of shape transitions in epitaxial islands, our in situ TEM observations shown here indicate that the supported Au clusters undergo the amorphalization–recrystallization oscillations as a dominant mechanism to respond to the interfacial strain effect.

## Discussion

The in situ HRTEM and HADDF imaging, EDS measurements, NEB and KMC simulations are mutually consistent and deliver strong evidence that Au atoms are separated from the Cu–Au alloy via adatom–substrate exchanges to result in the enrichment of Au adatoms at the surface due to the larger barrier for Au adatoms diffusing into the bulk than that for Cu adatoms. This phase separation process does not induce a net loss of the metal. This is different from the typical chemical dealloying process, where the less noble metal is chemically leached out from the parent alloy, leaving behind a residual porous structure[4–26]. The phase separation shown here requires high-temperature annealing of the sample that results in a large number of active sources (e.g., atomic steps, kinks, ledges) to form Cu and Au adatoms via step-edge detachments. Longer annealing time can lead to the formation of more Au clusters with a relatively larger average size until Au is nearly depleted from the parent phase. However, it is also worth mentioning that the thin film samples are prone to break during annealing due to thermal stress and fast surface mobility of adatoms in the vicinity of the void edges at the elevated temperatures. As a result, the area under the TEM investigation can be lost from prolonged annealing.

The Cu–Au alloys have the tendency to form ordered intermetallic compounds at the temperature up to ~390 °C. Our in situ TEM observations also confirmed the formation of an ordered $Cu_3Au$-like surface alloy by annealing Cu-10at.%Au films at ~350 °C (Supplementary Fig. 3). The resulting ordered surface alloy is induced by the interplay between the chemical ordering to form Cu–Au bonds and the tendency for surface segregation of Au atoms, where the latter favors the occupation of neighboring lattice sites by the same atomic species at the surface sites while chemical ordering causes exactly the opposite. The ordered Cu–Au alloy has improved surface stability and less tendency to undergo the phase separation because the pairwise atomic interaction results in the favored Cu–Au configuration. By contrast, the observed phase separation at 600 °C suggests that the pairwise Cu–Au atomic interaction is significantly weakened above the order–disorder transition temperature. The comparative observations made by annealing the Cu–Au films below and above the order–disorder transition temperature indicate that the tendency to the phase separation can be reduced by lowering the annealing temperature to promote Cu–Au pairwise interactions.

The observed phase separation process relies on the inherent asymmetric adatom–substrate exchange barriers to result in the enrichment of Au adatoms at the surface. We envision the broader applicability of this process because size differences between constituent atoms in multicomponent materials can typically lead to the different adatom–substrate exchange barriers between dissimilar atoms. However, stoichiometric, intermetallic compounds may have less tendency than solid solutions to undergo such a phase separation process because the strong pairwise interatomic interactions in intermetallic compounds may make the surface more stable at elevated temperatures, thereby reducing the number of active sources (e.g., atomic steps, kinks, and ledges) to form the fluid phase of adatoms. To further confirm this feature, we also performed scanning tunneling microscopy (STM) experiments by annealing intermetallic compound $Cu_3Au(100)$ at ~600 °C in UHV (Supplementary Note 8). As shown in Supplementary Fig. 12, the STM images indicate that the overall surface density of clusters is lower than that by annealing the Cu-10at.%Au(100) thin film (as shown in Fig. 1a) despite the higher Au content in the intermetallic $Cu_3Au$ crystal.

In conclusion, we have shown an unexpected phase separation process via the adatom–substrate exchange mechanism in a miscible alloy system. The disparity in the adatom–substrate exchange barriers results in the significant enrichment of solute atoms that subsequently aggregate as clusters. The clusters undergo rich dynamics of structure and shape evolution including grain rotation and amorphization–crystallization oscillations before becoming crystallographically aligned with the substrate. We envision the broader applicability of the results because of inherent asymmetric adatom-substrate exchange barriers for kinetically inducing the phase separation in multicomponent materials and the generality of the interface effect in modulating the dynamics of supported clusters.

## Methods

**Sample preparation and TEM characterization.** Our in situ TEM experiments were performed in a dedicated field-emission environmental TEM (FEI Titan 80-300) equipped with an objective lens aberration corrector and operated at 300 kV. Cu-10at.%Au(100) single-crystal thin films with ~500 Å thickness were grown on NaCl(100) by e-beam co-evaporation of Cu and Au, where the alloy composition was controlled by the evaporation rate of the two guns (Supplementary Fig. 1a). The Cu(Au) alloy films were transferred from the NaCl substrate by floatation in deionized water, washed, and mounted on the Dens Solutions Wildfire Nano-Chips and then loaded on a TEM specimen holder (Supplementary Fig. 1b). The as-prepared thin films had good continuity over large areas (Supplementary Fig. 2a) and uniform distribution of Cu and Au (Supplementary Fig. 2b, c). The quantification of the EDS spectra gives the atomic ratio of 90.7% Cu and 9.3% Au (Supplementary Fig. 2d, e). The Cu–Au films were annealed at ~350 °C in $H_2$ at a gas pressure of ~0.001 Torr to remove native oxide and generate faceted holes (Supplementary Fig. 3a). The complete removal of native oxide and the surface cleanliness were confirmed by electron diffraction (Supplementary Fig. 3b). EELS measurements were also performed to ensure that the thin films annealed in $H_2$ gas flow are free of oxygen (Supplementary Fig. 3c, d). In situ TEM observations of the phase separation were made in both the cross-sectional and planar views, where the cross-sectional TEM view was made through {100} and {110} facets (i.e., edges) of holes formed in the annealed Cu(Au) thin films. Notably, the in situ HRTEM imaging experiments were performed with thin film specimens at elevated temperature where significant atomic mobility and thermal drift can affect detrimentally the image contrast and resolution that can be achieved in practice.

**DFT calculations.** The DFT calculations were performed using the Vienna ab initio simulation package[57–60] with the PW91 generalized gradient approximation[61] and projector augmented wave[62] potentials with a cutoff energy of 600 eV. The Brillouin-zone integration was performed using $(4 \times 4 \times 1)$ K-point meshes based on Monkhorst–Pack grids[63] and with broadening of the Fermi surface according to Methfessel–Paxton smearing technique[64] with a smearing parameter of 0.2 eV. The surface was modeled by a periodically repeated slab consisting of four layers with the bottom two layers fixed, while the other layers were free to relax until all force components acting on the atoms are below 0.02 eVÅ$^{-1}$, and successive slabs are separated by a vacuum region of 12 Å. To calculate the energy barriers for the atomic hopping or exchanging, we applied the climbing image nudged elastic bands (CI-NEB) method[65], where we used three intermediate images between the initial and final states.

**KMC simulations.** The composition evolution in the fluid phase of Cu and Au adatoms induced by exchanges between the adatoms in the fluid phase and the Cu (Au) substrate surface atoms was simulated by KMC simulations[66]. The Cu(Au) (100) surface was modeled by a $100 \times 100$ supercell with 20,000 atoms. Two hundred adatoms were added onto the substrate surface to simulate the fluid composition evolution. The starting atomic composition of the substrate surface is assumed to be 50% Cu–50% Au, while the initial composition of the adatoms in the fluid phase is assumed to be 90% Cu–10% Au. During the simulations, the adatoms on the substrate surface are allowed to either hop on the substrate surface or exchange with an atom in the substrate surface. If an adatom diffuses out of the simulation cell, another adatom is added to the cell with 90% probability to be Cu and 10% probability to be Au, which is consistent with the initial composition of the fluid phase with the composition of 90% Cu–10% Au. The Au concentrations in the fluid phase and in the substrate surface were monitored throughout the surface exchange processes.

In our KMC simulations, all the possible hopping and exchanging paths of adatoms were considered for each step. The diffusive rate of each path was calculated by the harmonic transition state theory[67],

$$r_i = v_i \exp\left(-\frac{\Delta E_i}{k_B T}\right), \qquad (1)$$

where $\Delta E_i$ is the diffusion barrier of the path, which was obtained by the NEB calculations described above, and $v_i$ is the prefactor related to the vibrational frequency of atoms. It was found that the time evolution of our surface models is not sensitive to the value of $v_i$. Therefore, we simply set $v_i$ to be 10 THz for all paths. A hopping or exchanging process was set to happen at each step with the probability

$$p_i = \frac{r_i}{\sum_i r_i}. \qquad (2)$$

And the time of the system then evolved by

$$dt = \frac{1}{\sum_i r_i} \ln\left(\frac{1}{s}\right), \qquad (3)$$

where $s$ is a random number uniformly distributed between 0 and 1. The simulation was run for $10^6$ steps and the Au concentrations in the fluid phase and in the substrate surface were outputted every 1000 steps. The obtained results are shown in Fig. 2b, c. Supplementary Fig. 5 shows snapshots of the KMC simulated initial and final states of the substrate surface.

**MC and MD simulations.** We first equilibrated the Cu-10%Au substrate using hybrid MC + MD simulations[68] which allowed to relax the system. The simulations resulted in a thermodynamically relaxed system. The Cu–Au system contains 145,600 atoms with 10% Au and 90% Cu that are randomly mixed prior to the simulation. Interatomic interactions were modeled using the embedded atom method potential[69]. In each MC trial step, a randomly selected Au was swapped with another randomly selected Cu atom. The trial move was accepted with a probability within $\min\left(1, \exp\left(-\frac{\Delta U}{k_B T}\right)\right)$, where $\Delta U$ is the change in system energy and $T$ is temperature of 900 K. Following 100 MC trials, we performed 0.1 ps of MD simulation in the isothermal–isobaric condition (NPT ensemble). Following the procedure, the system was annealed with a total of 400,000 MC trails and 400 ps MD relaxation, which resulted in nonuniform Au distributions due to surface effects. The Au concentration on the surface increased from 10% to about

40% (see Supplementary Fig. 5). Then, we put a Au cluster on the equilibrated Cu–Au substrate followed by a long MD simulation of 1000 ps (1 ns). We simulated Au clusters with different misalignments with the substrate, where the different cluster orientations were created by cutting a cluster built with the Wulff construction. The Au clusters have about 12 layers of atoms, containing 7473, 6960, and 7495 atoms for the misalignment angles of 15°, 90°, and 0°, respectively. The MD simulations were performed in the NVT ensemble (constant atoms, volume, and temperature) at 900 K using a Nosé–Hoover thermostat[70]. We calculated the MSD of the atoms in the nanocluster as a function of time according to $\langle \Delta r^2(t) \rangle = \frac{1}{N} \sum_{i=1}^{N} [r_i(t + t_0) - r_i(t_0)]^2$, where $N$ is the total number of atoms in the supported Au cluster, $r_i(t_0)$ represents the initial position of atoms at time $t_0$, and $r_i(t + t_0)$ are the positions of atoms at time $t + t_0$.

## Data availability

All data generated or analysed during this study are included in this published article (and its Supplementary Information files).

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

## Acknowledgements

This work was supported by the U.S. Department of Energy, Office of Basic Energy Sciences, Division of Materials Sciences and Engineering under Award No. DE-SC0001135. This research used resources of the Center for Functional Nanomaterials and the Scientific Data and Computing Center, a component of the Computational Science Initiative, which is a U.S. DOE Office of Science Facility, at Brookhaven National Laboratory under Contract No. DE-SC0012704. This work used the computational resources from the Extreme Science and Engineering Discovery Environment (XSEDE) through allocation TG-DMR110009, which is supported by National Science Foundation grant number OCI-1053575. This research used resources of the Environmental TEM Catalysis Consortium (ECC), which is supported by the University of Pittsburgh and Hitachi High Technologies. STEM characterization was performed at the Nanoscale Fabrication and Characterization Facility (NFCF) in the Petersen Institute of NanoScience and Engineering (PINSE) at the University of Pittsburgh.

## Author contributions

G.Z. conceived the experiments and supervised the project. L.Z., D.Z., X.S., S.D.H., L.L., Q.Y., X.C., C.L., and H.Q. performed the experiments. P.C., Y.L., J.L., Y.Y., and G.W. conducted DFT, KMC, and NEB calculations. E.A.S. and J.C.Y. contributed new analytic tools. L.Z. and G.Z. analyzed data and wrote the paper. All the authors commented on the paper.

## Competing interests

The authors declare no competing interests.
