## [Peer Review File · Nature Communications]

Reviewers' comments:

Reviewer #1 (Remarks to the Author):

Comments:

In this manuscript, the authors reported an atomic-scale self-filtration process that results in the dealloying of a solid solution. The use of dynamic, atomic-scale electron microscopy observations and theoretical modeling to delineate the atomic-scale mechanisms is impressive. The texts and figures were well organized and logical, so I'd recommend to accept this manuscript after major revision.

1. During the high-temperature reaction, the Au clusters are amorphous-crystallization oscillations to adjust the crystal orientation due to the lattice mismatch of Cu, so whether the underlying Cu is also affected at the same time, resulting in a certain amorphization or a certain structural disorder, the final result is a two-phase dynamic adjustment?

2. If getting an ideal nanoporous structure, the prolonged dealloying duration is always required. Is the situation (dissolution, nucleation, crystallization, expansion, stacking, etc) within the initial surface dealloying period of Cu-Au similar to the stage of long-term dealloying? i.e., after 30 min or 60 min.

3. Is the initial crystallinity transformation during the dealloying to cluster universal or not? Because the dealloying-preferred alloy system is diverse and the dealloying phenomenon is always dependent of several factors such as alloy kind, composition ratio, phase content (single or multiple), corrosive potential gap between involved constituents, etc. For example, the dealloying-preferred alloy system generally contains continuous solid solution (Ag-Au) and fixed intermetallic compounds (CuAu, Cu₃Au, Au₃Cu, etc), which belongs to totally different counterpart and different re-crystallization rule during the selective leaching.

4. Why the surface step in Fig. S4 b isn't marked?

5. Minor mistakes:

1) Line 179, "the angle is marked by black and green lines in the inset diffractogram in Fig. 3e", the lines are missing.

2) The captions of Supplemental video 4 and 5 are incorrect, 4 is amorphous-to-crystalline while 5 is crystalline-to-amorphous transition.

3) Please check carefully about the format of references.

Reviewer #2 (Remarks to the Author):

In "Atomic-scale self-dealloying and clustering of solute atoms" the authors report on the surface reorganization of Cu-10at.%Au nanofoils annealed at 600°C in hydrogen. The kinetics of this process is studied using in-situ TEM and is corroborated by a combination of first-principle, kinetic Monte Carlo and molecular dynamics simulations.

Thermodynamics of small systems are a fascinating field as it often breaks with classical thermodynamics. Known phenomena at the nanoscale include mixing of macroscopically immiscible systems or interfacial sharpening at the interface of two fully miscible phases (Science 306, 1913 (2004), Appl. Phys. Lett. 99, 181902 (2011)).

Here, the authors use Cu-Au to study the possibility of phase separation in a miscible system.

Segregation, surface alloys and order-disorder transition in Cu-Au are not new and have been discussed extensively in the literature, see e.g.:

Phys. Rev. Lett. 51,1, 43-46 (1983)

Phys. Rev. B 45, 3703 (1992)

Phys. Rev. B 57, 6427 (1998)

Nanoscale, 2017, 9, 9267

Modelling Simul. Mater. Sci. Eng. 13 657 (2005)

Quite interestingly the authors have already published a very similar work on the same Cu-10%Au system, see:

Segregation induced order-disorder transition in Cu(Au) surface alloys (Acta Mater.154, p. 220-227 (2018)).

The authors should try to put their new work into context to what is known and stress clearly the difference between this and their previous work.

The manuscript needs significant improvement with respect to its clarity and description of the different processes at play. The authors should provide the reader a clear and comprehensive picture about the chemistry and kinetics in this system and compare with literature (e.g. Science 306, 1913 (2004)). I recommend major revisions including additional experiments before the manuscript is accepted.

Below I highlight some items I recommend revising:

1.) Chemical Analysis

The authors claim that the nanoparticles that form on the surface are made of gold. This claim clearly lacks experimental support. To distinguish between the two materials Au and Cu-10at.% the authors rely on the measurement of the local lattice spacing, which is not sufficient.

To substantiate their claim, the authors should perform detailed EDX mappings and line scans, aiming to visualize the chemical differences between the surface and the Cu-10at.%Au phase, see e.g. Nanoscale,2017, 9, 9267. Furthermore, the authors should provide an EDX spectrum of the as-deposited sample to confirm the initial composition.

2.) Which effect has the presence of the void on the observed cluster formation/"phase separation" and its kinetics? It is known that edges act as sources for capillary instabilities in thin films leading to enhanced surface diffusion in the vicinity of the edge, see e.g. (E. Jiran and C. V. Thompson, Capillary

instabilities in thin films, *J. Electron. Mater.*, 1990, 19, 1153–1160). The authors should carefully discuss which processes are responsible for the cluster formation.

3.) Hydrogen

Here, the samples have been annealed in hydrogen at elevated temperatures. It is known that the presence of hydrogen and solution in metals facilitates dislocation formation and increases their mobility (see e.g. *Hydrogen in Metals in Physical Metallurgy* (Elsevier)). This can induce grain rotation during recrystallization. The authors should discuss the role of hydrogen on the kinetics and observed recrystallization phenomena. Similar to their paper in *Acta Mater.*, the authors should also include the presence of hydrogen in their first-principle simulations.

4.) Sample preparation

The procedure of TEM sample preparation and imaging is not clear. The authors should add a schematic to the supplementary illustrating the different steps of sample preparation including the transfer on the heating chip.

Minor issues:

1. Please state the manufacturer of the heating chip that is used.
2. The scale bar in all diffractograms is missing.
3. Please report the acceleration voltage.
4. Please omit the word "self-filtration" and use established wording

Reviewer #3 (Remarks to the Author):

Taking the results and simulations at face value, this article certainly contains new, surprising (in places) and provocative information. As such, it satisfies the requirements to be published in a high-impact journal. Essentially, the authors find that the surface of a homogeneous metallic alloy can undergo a kind of vertical phase separation, wherein 3D clusters, and eventually crystallites, of Au emerge from the homogeneous bulk Cu-10%Au alloy. The maximum height is a surprising number of atom layers (>20), and goes well beyond ordinary surface segregation, or anything that is seen in STM studies that I am familiar with. All that is required to trigger this "phase transformation" is simple heating at 600 deg C in a low pressure of hydrogen.

There are a few topics that I would like to have seen discussed in more detail -

Is this phenomenon unique to thin alloy films of about the given thickness dimension (50 nm thick), or would it happen even on a bulk crystal surface?

Can the absence of a beam effect be quantified beyond "likely" (line 228)?

Is it possible that hydrogen plays some role?

How low is the oxygen content of the film (referring to bulk, not surface oxide)? Such oxygen could react with incoming hydrogen.

Although the exposure temperature is well above any order-disorder transition, is it possible that the type of pairwise atomic interaction implied by that ordering tendency plays some role in the phenomenon?

I am not really in agreement with calling this "dealloying", as that is bound to cause confusion, and the physics of this are quite different from what is normally understood by dealloying.

Reviewer #1 (Remarks to the Author):

In this manuscript, the authors reported an atomic-scale self-filtration process that results in the dealloying of a solid solution. The use of dynamic, atomic-scale electron microscopy observations and theoretical modeling to delineate the atomic-scale mechanisms is impressive. The texts and figures were well organized and logical, so I'd recommend to accept this manuscript after major revision.

Reply: We appreciate your careful reading and insightful suggestions and comments. We have made the revisions following your suggestions.

1. During the high-temperature reaction, the Au clusters are amorphous-crystallization oscillations to adjust the crystal orientation due to the lattice mismatch of Cu, so whether the underlying Cu is also affected at the same time, resulting in a certain amorphization or a certain structural disorder, the final result is a two-phase dynamic adjustment?

Reply: That is a great point! Our experimental setup provides cross-sectional views that enable dynamic observations of both supported Au and Cu substrate at the same time. As shown in the in-situ TEM image sequences (Fig. 4), our TEM observations indicate that the dynamic structural evolution in the Au clusters has no noticeable effect on the underlying Cu substrate. That is, the interfacial strain induced structure oscillations (amorphization and recrystallization) are limited to Au clusters. This is also confirmed from molecular dynamics simulations, which show structure oscillations in the supported Au clusters without noticeable changes in the Cu substrate (Fig. 5 in the manuscript). The higher stability of the Cu substrate over supported Au clusters can be attributed to the size effect, where the small Au clusters are more susceptible to interfacial strain fluctuations because of their large surface/interface area-to-volume ratio and lower barriers for the rearrangement of atoms compared to the Cu rigid substrate. This is also consistent with the large body of literature demonstrating that epitaxial islands typically undergo a series of shape transitions induced by interfacial strains whereas the substrate remains unaffected. Therefore, the interfacial strain effect is dominated by dynamic structure evolution in the supported clusters.

We have incorporated this point into the revision as follows.

“The in-situ TEM observations in Figs. 3 and 4 show that the dynamic structural evolution in the Au clusters has no noticeable effect on the underlying Cu substrate. That is, the interfacial strain induced structure oscillations (amorphization and recrystallization) are limited to Au clusters. This is also confirmed from molecular dynamics simulations (Fig. 5), which show that structure oscillations in the supported Au clusters do not induce noticeable structure changes in the Cu substrate. The higher stability of the substrate over supported Au clusters can be attributed to the size effect, where the small Au clusters are more susceptible to interfacial strain fluctuations because of their large surface/interface area-to-volume ratio and lower barriers for the rearrangement of atoms compared to the rigid substrate. This is also consistent with the large body of literature demonstrating that epitaxial islands typically undergo a series of shape transitions induced by interfacial strains whereas the substrate remains relatively unaffected [1-4]. Instead of shape transitions in epitaxial islands, our in-situ TEM observations shown here indicate that the supported Au clusters undergo the amorphization-recrystallization oscillations as a dominant mechanism to respond to the interfacial strain effect.” (please see page 11, lines 3-16)

References

- 1 Brongersma, S., Castell, M., Perovic, D. & Zinke-Allmang, M. Stress-induced shape transition of CoSi₂ clusters on Si (100). *Physical review letters* **80**, 3795 (1998).
- 2 Daruka, I., Grossauer, C., Springholz, G. & Tersoff, J. Equilibrium phase diagrams for the elongation of epitaxial quantum dots into hut-shaped clusters and quantum wires. *Physical Review B* **89**, 235427 (2014).
- 3 Tersoff, J. & Tromp, R. Shape transition in growth of strained islands: spontaneous formation of quantum wires. *Physical review letters* **70**, 2782 (1993).
- 4 Zhou, G. & Yang, J. C. Formation of Quasi-One-Dimensional Cu₂O Structures by in situ Oxidation of Cu(100). *Physical review letters* **89**, 106101 (2002).

2.If getting an ideal nanoporous structure, the prolonged dealloying duration is always required. Is the situation (dissolution, nucleation, crystallization, expansion, stacking, etc) within the initial surface dealloying period of Cu-Au similar to the stage of long-term dealloying? i.e., after 30 min or 60 min.

Reply: Thank you for this question. First, we would like to clarify that the phase separation process reported in this manuscript is completely different from the chemical dealloying mechanism. In our case, Au atoms are separated from the Cu-Au alloy via a physical process of adatom-substrate exchanges, which results in the enrichment of Au adatoms at the surface due to the larger barrier for Au adatoms diffusing into the bulk than that for Cu adatoms. This process does not result in a nanoporous structure because there is no net loss of the metal from the alloy. This is different from the typical chemical dealloying, where the less noble metal is chemically leached out from the parent alloy, leaving behind a residual porous structure.

Please also note that the phase separation reported here requires high-temperature annealing of the sample that results in a large number of active sources (e.g., atomic steps, kinks, ledges) to form Cu and Au adatoms via step-edge detachments. Longer annealing time leads to the formation of more Au clusters with a relatively larger average size until Au is nearly depleted from the parent phase. It is also worth mentioning that the thin film samples are prone to break during heating due to thermal stress and fast surface mobility of adatoms in the vicinity of void edges at the elevated temperatures. As a result, the area under the TEM investigation can be lost from prolonged annealing.

We have clarified this point in the revision as follows:

“The combination of HRTEM and HADDF imaging, EDS, NEB and KMC simulations is mutually consistent and delivers strong evidence that Au atoms are separated from the Cu-Au alloy via the physical process of adatom-substrate exchanges that results in the enrichment of Au adatoms at the surface due to the larger barrier for Au adatoms diffusing into the bulk than that for Cu adatoms. This phase separation process does not result in a nanoporous structure in the parent alloy because there is no net loss of the metal. This is different from the typical chemical dealloying, where the less noble metal is chemically leached out from the parent alloy, leaving behind a residual porous structure. The phase separation shown here requires high-temperature annealing of the sample that results in a large number of active sources (e.g., atomic steps, kinks, ledges) to form Cu and Au adatoms via step-edge detachments. Longer annealing time leads to the formation of more Au clusters with a relatively larger average size until Au is nearly depleted from the parent phase. It is also worth mentioning that the thin film samples are prone to break during annealing due to thermal stress and fast surface mobility of adatoms in the vicinity of the void edges at the elevated temperatures. As a result, the area

under the TEM investigation can be lost from prolonged annealing.” (please see page 11, lines 13-27)

3. Is the initial crystallinity transformation during the dealloying to cluster universal or not? Because the dealloying-preferred alloy system is diverse and the dealloying phenomenon is always dependent of several factors such as alloy kind, composition ratio, phase content (single or multiple), corrosive potential gap between involved constituents, etc. For example, the dealloying-preferred alloy system generally contains continuous solid solution (Ag-Au) and fixed intermetallic compounds (CuAu, Cu₃Au, Au₃Cu, etc), which belongs to totally different counterpart and different re-crystallization rule during the selective leaching.

Reply: Thank you for this question. As described in our response above, the observed surface segregation process relies on the inherent asymmetric adatom-substrate exchange barriers to result in the enrichment of Au atoms at the surface. We envision the broader applicability of this phase separation process because size differences between constituent atoms in alloys can typically result in different adatom-substrate exchange barriers between dissimilar atoms. However, stoichiometric intermetallic compounds may have less tendency than solid solutions to undergo such a dealloying process because the strong pairwise interatomic bonding in intermetallic compounds can make the surface more stable at the elevated temperature, thereby reducing the number of active sources (e.g., atomic steps, kinks, ledges) to form adatoms via step-edge detachments. This is consistent with our new STM experiments performed on the intermetallic compound Cu₃Au(100) surface, where the formation of Au clusters is still observed from the ultrahigh vacuum (UHV) annealing of the intermetallic Cu₃Au, but their surface density is lower than that for Cu-10at.%Au(100) thin films despite the higher Au content in the intermetallic Cu₃Au crystal.

Supplementary Figure 12: STM images obtained from the Cu₃Au(100) annealed at ~ 600°C in ultrahigh vacuum. (a) A typical surface area that is nearly free of clusters, (b) a separate surface area showing the presence of clusters.

We have incorporated this comment into the revision as follows:

“The observed phase separation process relies on the inherent asymmetric adatom-substrate exchange barriers to result in the enrichment of Au atoms at the surface. We envision the broader applicability of this process because size differences between constituent atoms in alloys can typically result in the different adatom-substrate exchange barriers between dissimilar atoms. However, stoichiometric, intermetallic compounds may have less tendency than solid solutions to undergo such a phase separation process because the strong interatomic bonding in intermetallic compounds can make the surface more stable at the elevated temperature, thereby reducing the number of active sources (e.g., atomic steps, kinks, ledges) to form adatoms via step-edge detachments. To further confirm this feature, we also performed scanning tunneling microscopy (STM) experiments by annealing intermetallic compound $\text{Cu}_3\text{Au}(100)$ at $\sim 600^\circ\text{C}$ in ultrahigh vacuum. As shown in Supplementary Fig. 12, the STM images indicate that the overall density of the clusters is lower than that on the $\text{Cu-10at.\%Au}(100)$ thin films (as shown in Fig. 1(a)) despite the higher Au content in the intermetallic Cu_3Au crystal.” (please see page 12, lines 11-23)

We have also included the STM images obtained from $\text{Cu}_3\text{Au}(100)$ into the supplementary information (Supplementary Fig. 12).

4. Why the surface step in Fig. S4 b isn't marked?

Reply: The in-situ TEM images in Fig. S4 illustrate the fast decay of steps at the elevated temperature. The surface step is still visible in Fig. S4b but the image had reduced contrast due to thermal drift of the specimen. As shown below, we replaced this image with another one obtained at a different moment of the time sequence that had less thermal drift and thus better image contrast to resolve the surface step.

Supplementary Figure 6: Surface steps act as active sources of Cu and Au adatoms. a-d, *In situ* TEM observations showing the retraction motion of surface steps during the annealing of a Cu-10at.%Au(100) film at 600°C and 1×10^{-3} Torr of H₂ gas flow, which results in a flux of Cu and Au adatoms on the planar surface.

5.Minor mistakes:

1) Line 179, “the angle is marked by black and green lines in the inset diffractogram in Fig. 3e”, the lines are missing.

Reply: Thank you for pointing out this error. The misalignment angle between the two spots is too small, putting two lines may block the diffraction spots. Therefore, we choose to mark the misalignment angle of the two diffraction spots using two circles. We have corrected the description as follows:

“the angle is determined by the measured clockwise rotation of the diffraction spots marked by blue and red rings in the inset diffractogram in Fig. 4e.”

2) The captions of Supplemental video 4 and 5 are incorrect, 4 is amorphous-to-crystalline while 5 is crystalline-to-amorphous transition.

Reply: Thank you for the careful reading, we have corrected the error in the revised version.

3) Please check carefully about the format of references.

Reply: We have corrected the references to fit for the Nature Communications format.

Reviewer #2 (Remarks to the Author):

In “Atomic-scale self-dealloying and clustering of solute atoms” the authors report on the surface reorganization of Cu-10at.%Au nanofoils annealed at 600°C in hydrogen. The kinetics of this process is studied using in-situ TEM and is corroborated by a combination of first-principle, kinetic Monte Carlo and molecular dynamics simulations.

Thermodynamics of small systems are a fascinating field as it often breaks with classical thermodynamics. Known phenomena at the nanoscale include mixing of macroscopically immiscible systems or interfacial sharpening at the interface of two fully miscible phases (Science 306, 1913 (2004), Appl. Phys. Lett. 99, 181902 (2011)).

Here, the authors use Cu-Au to study the possibility of phase separation in a miscible system. Segregation, surface alloys and order-disorder transition in Cu-Au are not new and have been discussed extensively in the literature, see e.g.: Phys. Rev. Lett. 51,1, 43-46 (1983), Phys. Rev. B 45, 3703 (1992) Phys. Rev. B 57, 6427 (1998) Nanoscale, 2017, 9, 9267 Modelling Simul. Mater. Sci. Eng. 13 657 (2005).

Quite interestingly the authors have already published a very similar work on the same Cu-10%Au system, see: Segregation induced order-disorder transition in Cu(Au) surface alloys (Acta Mater. 154, p. 220-227 (2018)). The authors should try to put their new work into context to what is known and stress clearly the difference between this and their previous work.

The manuscript needs significant improvement with respect to its clarity and description of the different processes at play. The authors should provide the reader a clear and comprehensive picture about the chemistry and kinetics in this system and compare with literature (e.g. Science 306, 1913 (2004)). I recommend major revisions including additional experiments before the manuscript is accepted.

Reply: We appreciate your assessment of our results. We also thank you for taking time and energy to offer us insightful comments and suggestions, which we found are very useful as we approached our revision.

First of all, we have followed your suggestion and added the following sentences to clarify the difference between the present work and our previous work:

“The Cu-Au system develops several intermetallic compounds that are stable to temperatures of ~ 390°C. Our previous in-situ TEM observations have shown that annealing Cu-10at.%Au solid solutions below the order-disorder transition temperature results in the formation of an ordered Cu₃Au-like surface alloy [1]. The resulting ordered surface alloy is found to have a profound effect on surface properties including acting as an effective barrier to inhibit dislocation annihilation at free surfaces [2] and slow down surface oxidation [3]. By contrast, here we demonstrate an unexpected phase separation phenomenon by simply annealing the Cu-10at.%Au solid solutions above the order-disorder transition temperature that results in the massive formation of Au clusters.”

References

- 1 Zou, L. *et al.* Atomically visualizing elemental segregation-induced surface alloying and restructuring. *The journal of physical chemistry letters* **8**, 6035-6040 (2017).

- 2 Zou, L. *et al.* Dislocation nucleation facilitated by atomic segregation. *Nature materials* **17**, 56-63 (2018).
- 3 Zou, L. *et al.* Segregation induced order-disorder transition in Cu (Au) surface alloys. *Acta Materialia* **154**, 220-227 (2018).

As described below, we have also performed additional experiments to address your other concerns and comments.

Below I highlight some items I recommend revising:

1.) Chemical Analysis

The authors claim that the nanoparticles that form on the surface are made of gold. This claim clearly lacks experimental support. To distinguish between the two materials Au and Cu-10at.% the authors rely on the measurement of the local lattice spacing, which is not sufficient. To substantiate their claim, the authors should perform STEM-HAADF experiments and detailed EDX mappings and line scans, aiming to visualize the chemical differences between the surface and the Cu-10at.%Au phase, see e.g. *Nanoscale*, 2017, 9, 9267. Furthermore, the authors should provide an EDX spectrum of the as-deposited sample to confirm the initial composition.

Reply: We are very appreciative of these comments. We have followed your suggestion and performed STEM-HAADF and EDS mapping/line scans to confirm the chemical composition. The results are given in Supplementary Fig. 2 and also shown below. As shown by the low-magnification HAADF image (Supplementary Fig. 2a, the as-prepared Cu-Au film shows good continuity. EDS elemental mapping and spectral analyses showed that the as-prepared Cu-Au film has the uniform composition of Cu and Au (Supplementary Figs. 2b-c. When obtaining the EDS spectra, the electron beam was spread to cover a large area to yield a statistically averaged results. The table (Supplemental Figs. 2d-e) presents the results of the quantitative analysis of the EDS spectra obtained from the as-prepared Cu-Au film, showing an atomic ratio of 90.7%Cu and 9.3%Au. The EDS measured composition is very close to the targeted composition of the Cu-10at.%Au film made by e-beam co-evaporation of Cu and Au, where the alloy composition was controlled by the evaporation rate of the two e-guns.

We have incorporated the HAADF and EDS results from the as-prepared Cu-Au films into the supplementary material (Supplementary Note 1 and Supplementary Fig. 2). In addition, the following sentences are also added into the main text:

“The as-prepared Cu-Au films have good continuity with uniform distribution of Cu and Au (Supplementary Fig. 2). Quantitative analysis of the EDS spectra obtained from the as-prepared Cu-Au film showed an atomic ratio of 90.7%Cu and 9.3%Au, which is very close to the targeted composition by controlling the evaporation rate of the two electron guns.” (please see page 3, lines 7-10)

Supplementary Fig. 2: HAADF and EDS analysis of the as-prepared Cu-10at.%Au(100) thin films. (a) Low-magnification HAADF image showing the good film continuity over the large area. (b, c) STEM-EDS mapping showing the uniform distribution of Cu and Au, respectively. (d) Representative EDS spectrum of the as-prepared film. (e) Chemical composition by quantification of EDS data, showing an atomic ratio of 90.7%Cu and 9.3%Au, which is very close to the targeted composition of the Cu-10at.%Au film made by e-beam co-evaporation of Cu and Au, where the alloy composition was controlled by manipulating the evaporation rate of Cu and Au.

We have also performed HAADF image and EDS line scan analysis on annealed Cu-Au films and the results are incorporated into the main text as in Fig. 2 and also given below. As marked by the dashed lines in the low-magnification HAADF-STEM image (Fig. 2A), clusters are visible along the edge of a void (the void is formed by thermal stress during annealing the film at 600°C), consistent with the E-TEM observations as shown in Fig. 1A. Inset in Fig. 2(A) is an atomically resolved HAADF-STEM image obtained from the cluster as marked by the dashed box in Fig. 2(A), in which the measured lattice spacings match well with the interplanar spaces of Au(111) and (110). Meanwhile, the EDS linescan across the cluster as marked by the dashed arrow in Fig. 2A indicates that the cluster consists of 88% of Au and 12%Cu (atomic percentage). Because our ETEM does not have the capability to perform EDS measurements, the annealed sample had to be cooled down and then transferred to a dedicated STEM for HAADF and EDS analyses. The specimen cooling and transfer may induce slight changes to the specimen, but the measured high Au content (88%) confirms that the segregated clusters are dominated by Au atoms, consistent with the measured crystal lattice spacings in the HRTEM images (Fig. 1(B, C)) and the HAADF image (inset in Fig. 2(A)) as well as the prediction from our KMC simulations (Fig. 1(f)).

Fig. 1(d, e): (d) Low-magnification HAADF-STEM image showing the formation of Au clusters (as marked by dashed lines) along a hole edge. The enlarged view is an atomically resolved HAADF-STEM image obtained from the cluster marked by the blue rectangle box. (e) STEM-EDS linescan along the arrow indicated in (d), confirming the high content of Au in the cluster.

We have incorporated these HAADF and EDS results into the main text (Fig. 1(d, e)), and added the following passage into the manuscript:

“The formation of Au clusters is further confirmed by high-angle annular dark-field scanning transmission electron microscopy (HAADF-STEM) imaging and energy dispersive X-ray spectroscopy (EDS) composition analysis. As marked by the dashed lines in the low-magnification HAADF-STEM image (Fig. 1d), clusters are visible along the edge of a void, consistent with the E-TEM observations (Fig. 1a). The inset in Fig. 1(d) is an atomically resolved HAADF-STEM image obtained from the cluster marked by the dashed box, in which the measured lattice spacings match well with the interplanar spacings of Au(111) and (110). Meanwhile, the EDS linescan across the cluster as marked by the dashed arrow in Fig. 1(d) indicates that the cluster consists of 88% of Au and 12%Cu (atomic percentage). The measured high Au content confirms that the segregated clusters are dominated by Au atoms, consistent with the measured crystal lattice spacings in the HRTEM images (Fig. 1(b, c)) and the HAADF image (inset in Fig. 1(d)) as well as the prediction from our KMC simulations shown below in Fig. 2.” (please see page 3, lines 30-31 and page 4, lines 1-11)

2.) Which effect has the presence of the void on the observed cluster formation/”phase separation” and its kinetics? It is know that edges act as sources for capillary instabilities in thin films leading to enhanced surface diffusion in the vicinity of the edge, see e.g. (E. Jiran and C. V. Thompson, Capillary instabilities in thin films, J. Electron. Mater., 1990, 19, 1153–1160). The authors should carefully discuss which processes are responsible for the cluster formation.

Reply: This is a great point that indeed requires elaboration. As mentioned earlier, the formation of voids in the Cu-Au films is induced by thermal stress during annealing. At the elevated temperature (600°C), these voids can grow larger due to the surface curvature effect on enhancing surface diffusion in vicinity of the void edge. The void growth results in a large number of mobile Cu and Au adatoms on the surface. Therefore, the edges of voids act a source of Cu and Au adatoms. In addition, surface steps are another source of Cu and Au adatoms. As shown in Supplementary Fig. 7, the fast decay (i.e., retraction motion) of surface steps results in a large number of Cu and Au adatoms to form a Cu-Au fluid phase. The

subsequent adatom-substrate exchanges results in the enrichment of Au adatoms in the fluid phase because of the larger barriers for the incorporation of Au adatoms into the substrate (as confirmed from our NEB and KMC simulations in Fig. 1(d-f)). The aggregation of Au adatoms in the fluid phase results in the formation of Au clusters. As shown in Fig. 1(a), Au adatoms can aggregate to form clusters on the planar surface area and along the edges of the voids. Our in-situ TEM observations are mainly focused on Au clusters along void edges because the edge-on, cross-sectional imaging allows for simultaneously visualizing structural evolution in both the cluster and the Cu substrate as well as their interfacial interactions.

In addition, we would like to clarify the annealing temperature effect. Our in-situ TEM observations indicate that the annealing of Cu-Au films at a lower temperature (e.g., 350°C) does not induce noticeable formation of Au clusters.

Supplementary Figure 3: TEM characterization of Cu-10at.%A(100) film annealed at ~ 350 °C and ~ 0.001 Torr of H₂ gas flow. (a) A representative faceted hole formed in the annealed film, the side facets are typically composed of {100} and {110} surface terminations. (b) Electron diffraction pattern along the [001] zone axis, displaying the single crystalline feature of the film, the absence of additional spots confirms the complete removal of native oxide by annealing in H₂ gas flow. (c, d) EELS O-K edge and Cu-L_{2,3} edges showing the presence of oxygen in the thin film before annealing and the absence of oxygen after annealing. The black and red ones are obtained from the unannealed and annealed sample, respectively. (e, d) HRTEM images showing the formation of an ordered Cu₃Au-like surface alloy along the (100) and (110) side facets as shown in (a).

According to the Cu-Au equilibrium phase diagram, the ordered phases of Cu-Au intermetallic compounds are stable to temperatures of ~ 390°C. Indeed, our in-situ TEM observations

showed that an ordered Cu₃Au-like surface alloy develops along void edges for annealing the Cu-10at.%Au film at ~ 350°C. As illustrated above in Supplementary Fig. 3, the HRTEM images show that the void edges are highly faceted, and both the (100) and (110) edges develop into a half-unit-cell thin layer of the Cu₃Au surface alloys, where the alternate bright and dark contrast of atom columns confirms the formation of an ordered Cu-Au surface alloy. The ordered Cu-Au surface alloy has improved surface stability and thus less tendency to form Au clusters. This is evident as shown in the TEM image of Supplementary Fig. 3a, where both the void edges and adjacent planar surface area are free of Au clusters.

In the revision, we have included the above TEM results (Supplementary Fig. 3) obtained from the Cu-10at.%Au(100) film annealed at 350°C in supplementary information. In addition, the following sentences are incorporated into the main text to clarify the effect of void edge and annealing temperature on the formation of Au clusters:

“In addition, the thermally induced voids in the Cu-Au films can grow larger due to the surface curvature effect on enhancing surface diffusion in the vicinity of the void edge. Therefore, the edges of voids also act as an active source of Cu and Au adatoms to join the fluid phase of Cu and Au adatoms on the surface.” (please see page 4, lines 27-30)

“The Cu-Au alloys have the tendency to form ordered intermetallic compounds at the temperature up to ~ 390°C. Our in-situ TEM observations also confirmed the formation of an ordered Cu₃Au-like surface alloy by annealing Cu-10at.%Au films at ~ 350°C (Supplementary Fig. 3). The resulting ordered surface alloy is induced by the interplay between the chemical ordering to form Cu-Au bonds and the tendency for surface segregation of Au atoms, where the latter favors the occupation of neighboring lattice sites by the same atomic species at the surface sites while chemical ordering causes exactly the opposite. The ordered Cu-Au alloy has improved surface stability and less tendency to undergo the phase separation because the pairwise atomic interaction results in the favored Cu-Au configuration. By contrast, the observed phase separation at 600°C suggests that the pairwise Cu-Au atomic interaction is significantly weakened above the order-disorder transition temperature. The comparative observations made by annealing the Cu-Au films below and above the order-disorder transition temperature indicate that the tendency to the phase separation can be reduced by lowering the annealing temperature to promote Cu-Au pairwise interactions.” (please see page 12, lines 5-18)

3.) Hydrogen

Here, the samples have been annealed in hydrogen at elevated temperatures. It is known that the presence of hydrogen and solution in metals facilitates dislocation formation and increases their mobility (see e.g. Hydrogen in Metals in Physical Metallurgy (Elsevier)). This can induce grain rotation during recrystallization. The authors should discuss the role of hydrogen on the kinetics and observed recrystallization phenomena. Similar to their paper in Acta Mater., the authors should also include the presence of hydrogen in their first-principle simulations.

Reply: Thank you for raising this point, which we agree indeed requires elaboration. The purpose of using hydrogen is to provide a reducing environment to remove any native oxide and maintain the surface cleanliness. This is confirmed by EELS measurements showing the absence of oxygen signal from the Cu-Au film annealed in H₂ (please see Supplementary Fig. 3c). This is also consistent with our ambient-pressure X-ray photoelectron spectroscopy (AP-XPS) measurements (1), which showed that any bulk-dissolved oxygen in Cu can be completely deoxygenated by flowing H₂ gas at ~580°C to form H₂O molecules that spontaneously desorb from the surface, resulting in an atomically clean Cu surface at the elevated temperature. To

further confirm whether the H₂ gas has any effect on the surface segregation, we recently employed an ambient-pressure X-ray photoelectron spectroscopy (AP-XPS) system to monitor the surface composition evolution of Cu₃Au(100) during annealing between under ultrahigh vacuum (UHV) and 0.1 Torr of H₂ gas flow (2). As shown below, no noticeable differences in the surface composition can be observed between the UHV and H₂ gas flow. This is consistent with other studies showing the high dissociation barriers of H₂ molecules on both Cu and Au surfaces [3-5]. Even for atomic hydrogen, it bonds weakly to Cu and Au, and desorbs from Au surfaces at the temperature of above ~ -163°C [6] and from Cu surfaces at the temperature of ~88°C [7]. Our in-situ TEM experiments were performed at ~ 600°C, much higher than the desorption temperatures for atomic hydrogen on Cu and Au surfaces.

Supplementary Fig. 11. Comparison in surface Au composition of Cu₃Au(100) for the annealing at 600°C between ultrahigh vacuum (UHV) (a) and 0.1 Torr of H₂ gas flow (b). Spectra were taken with the photon energy of 400 eV. The Au 4f region consists of two contributions corresponding to Au-4f_{7/2} and Au-4f_{5/2}, respectively. Both contributions can be deconvoluted into two components, i.e., pure Au (Au_p) and alloyed Au (Au_a). The annealing under the H₂ gas flow does not induce any notable differences in the pure Au concentration, as indicated by the Au_a/Au_p ratio of integrated peak areas of each Au species.

In the revision, we have clarified this point by including the AP-XPS measurements in the supplementary information (Supplementary Note 10) and incorporating the following sentences into the main text:

“In addition, the H₂ gas flow has negligible influence on the observed phase segregation other than providing a reducing environment to maintain the surface cleanliness. This is confirmed by EELS measurements showing the absence of oxygen signal from the Cu-Au film annealed in H₂ (Supplementary Fig. 11). This is also consistent with our ambient-pressure X-ray photoelectron spectroscopy (AP-XPS) measurements (1), which showed that any bulk-dissolved oxygen in Cu can be completely deoxygenated by flowing H₂ gas at ~580°C to form H₂O molecules that spontaneously desorb from the surface, resulting in an atomically clean Cu surface at the elevated temperature [1]. To further confirm whether the H₂ gas has any effect on the surface segregation, we also employed AP-XPS to monitor the surface composition evolution of Cu₃Au(100) during annealing under ultrahigh vacuum (UHV) and in 0.1 Torr of H₂ gas flow [2]. No noticeable differences in the surface composition can be observed between the UHV and H₂ gas flow (Supplementary Fig. 11). This is consistent with other studies showing the high dissociation barriers of H₂ molecules on both Cu and Au surfaces [3-5]. Even for atomic hydrogen, it bonds weakly to Cu and Au, and desorbs from Au surfaces at the temperature above ~ -163°C [6] and from Cu surfaces at the temperature of ~88°C [7], both of which are much lower than the annealing temperature in our in-situ experiments..” (please see page 9, lines 24-31 and page 10, lines 1-9)

References:

1. C.R. Li, P.H. Zhang, J.Y. Wang, J.A. Boscoboinik, G.W. Zhou, “Tuning the Deoxygenation of Bulk-Dissolved Oxygen in Copper”, *Journal of Physical Chemistry C* 122, 8254-8261(2018)
2. C. R. Li, Q.Q. Liu, J. A. Boscoboinik, and G. W. Zhou, Tuning the surface composition of Cu₃Au binary alloy, *Phys. Chem. Chem. Phys.* 2020,22, 3379-3389
3. B. Hammer and J. K. Norskov, Why gold is the noblest of all the metals, *Nature*, 1995, 376, 238–240.
4. L. Stobiński and R. Duś, Model of atomic hydrogen adsorption on thin gold film surface, *Vacuum*, 1994, 45, 299–301.
5. J. Greeley and M. Mavrikakis, Surface and Subsurface Hydrogen: Adsorption Properties on Transition Metals and Near-Surface Alloys, *J. Phys. Chem. B*, 2005, 109, 3460–3471.
6. M. Pan, A. J. Brush, Z. D. Pozun, H. C. Ham, W.-Y. Yu, G. Henkelman, G. S. Hwang and C. B. Mullins, Model studies of heterogeneous catalytic hydrogenation reactions with gold, *Chem. Soc. Rev.*, 2013, 42, 5002–5013.
7. I. Yasumori, N. Momma and M. Kiyomiya, Mechanism of Hydrogen Adsorption and Hydrogen-Deuterium Equilibration on Copper Surface, *Jpn. J. Appl. Phys.*, 1974, 13, 485.

4.) Sample preparation

The procedure of TEM sample preparation and imaging is not clear. The authors should add a schematic to the supplementary illustrating the difference steps of sample preparation including the transfer on the heating chip.

Reply: Thank you for this suggestion. We have followed your suggestion and added the following schematic to the supplementary material to illustrate the procedure of preparing the Cu-Au thin films and subsequent steps in transferring the film to the heating chip.

Supplementary Fig. 1: (a) Schematic of the electron-beam evaporation of Cu-Au thin films on NiAl(100), where the film composition is controlled by manipulating the evaporation rate of Cu and Au in the crucibles. (b) The as-prepared Cu-10at.%Au film is then removed from the NaAl substrate by dissolution of the NaCl in deionized water, washed, and mounted on a TEM specimen holder.

Minor issues:

- 1. Please state the manufacturer of the heating chip that is used.**

Reply: The heating used for the experiments is Dens solutions Wildfire Nano-Chips. We have added the information in the updated manuscript.

- 2. The scale bar in all diffractograms is missing.**

Reply: We have added scale bar to the electron diffraction pattern to Supplementary Figure 1 All the other diffractograms are obtained from the FFT operation of the HRTEM images, the scale bar information is given in the HRTEM images, the scale bars are not shown in the figure.

- 3. Please report the acceleration voltage.**

Reply: We have added the acceleration voltage (300 keV) in the experimental section.

- 4. Please omit the word “self-filtration” and use established wording**

Reply: Thank you for this suggestion, we have replaced the word “self-filtration” with “phase separation”

Reviewer #3 (Remarks to the Author):

Taking the results and simulations at face value, this article certainly contains new, surprising (in places) and provocative information. As such, it satisfies the requirements to be published in a high-impact journal. Essentially, the authors find that the surface of a homogeneous metallic alloy can undergo a kind of vertical phase separation, wherein 3D clusters, and eventually crystallites, of Au emerge from the homogeneous bulk Cu-10%Au alloy. The maximum height is a surprising number of atom layers (>20), and goes well beyond ordinary surface segregation, or anything that is seen in STM studies that I am familiar with. All that is required to trigger this "phase transformation" is simple heating at 600 deg C in a low pressure of hydrogen.

Reply: We greatly appreciate your encouraging comments and insightful assessment of our results. We are also grateful for the time and energy you expended on our behalf.

1. There are a few topics that I would like to have seen discussed in more detail - Is this phenomenon unique to thin alloy films of about the given thickness dimension (50 nm thick), or would it happen even on a bulk crystal surface?

Reply: This is a great point that made us think a great deal about our results. As described in the manuscript, the observed phase separation relies on the existence of active sources (e.g., step edges, kinks, and void edges) to form a large number of Cu and Au adatoms. At annealing temperature of 600°C, the surface has a large number of such active sites to result in massive Cu and Au adatoms via the fast retraction motion of surface steps (e.g., Supplementary Fig. 7) and the growth of thermal voids.

Supplementary Figure 3: TEM characterization of Cu-10at.%A(100) film annealed at ~ 350 °C and ~ 0.001 Torr of H₂ gas flow. (a) A representative faceted hole formed in the annealed film, the side facets are typically composed of {100} and {110} surface terminations. (b) Electron diffraction pattern along the [001] zone axis, displaying the single crystalline feature of the film, the absence of additional spots confirms the complete removal of native oxide by annealing in

H₂ gas flow. (c, d) EELS O-K edge and Cu-L_{2,3} edges showing the presence of oxygen in the thin film before annealing and the absence of oxygen after annealing. The black and red ones are obtained from the unannealed and annealed sample, respectively. (e, d) HRTEM images showing the formation of an ordered Cu₃Au-like surface alloy along the (100) and (110) side facets as shown in (a).

According to the Cu-Au equilibrium phase diagram, the order-disorder transition temperature for Cu-Au intermetallic compounds is ~ 390°C. As shown in the figure above (i.e., Supplementary Fig. 3), our in-situ TEM observations showed that an ordered Cu₃Au-like surface alloy develops along void edges for annealing Cu-10at.%Au films at ~ 350°C. The HRTEM images show that the void edges are highly faceted, and a half-unit-cell thin layer of the Cu₃Au surface alloy is formed along the (100) and (110) edges, where the alternate bright and dark contrast of atom columns confirms the formation of an ordered Cu-Au surface alloy. The ordered Cu-Au surface alloy has improved surface stability and thus less tendency to form Au clusters. This is evident from the TEM image above (Supplementary Fig. 3a), where both the edges of the void and adjacent planar surface area are free of Au clusters.

We have added the following sentences to clarify this point:

“The Cu-Au alloys have the tendency to form ordered intermetallic compounds at the temperature up to ~ 390°C. Our in-situ TEM observations also confirmed the formation of an ordered Cu₃Au-like surface alloy by annealing Cu-10at.%Au films at ~ 350°C (Supplementary Fig. 3). The resulting ordered surface alloy is induced by the interplay between the chemical ordering to form Cu-Au bonds and the tendency for surface segregation of Au atoms, where the latter favors the occupation of neighboring lattice sites by the same atomic species at the surface sites while chemical ordering causes exactly the opposite. The ordered Cu-Au alloy has improved surface stability and less tendency to undergo the phase separation because the pairwise atomic interaction results in the favored Cu-Au configuration. By contrast, the observed phase separation at 600°C suggests that the pairwise Cu-Au atomic interaction is significantly weakened above the order-disorder transition. The comparative observations made by annealing the Cu-Au films below and above the order-disorder transition temperature indicate that the tendency to the phase separation can be reduced by lowering the annealing temperature to promote Cu-Au pairwise interactions.” (please see page 12, lines 3-16)

To further confirm that the stoichiometric intermetallic compounds have less tendency to undergo the phase separation, we performed additional scanning tunneling microscopy (STM) experiments by annealing a single-crystal Cu₃Au(100) at ~ 600°C in ultrahigh vacuum (UHV). As illustrated below, the STM images show that most of the surface area is free of clusters with the presence of clusters in some scattered areas. Overall, the surface density of the clusters is much lower than that formed on the Cu-10%Au(100) film under the similar annealing condition, despite the higher content of Au in the intermetallic compound. These STM results confirm that the intermetallic compound has improved surface stability and therefore is less likely to develop the phase separation.

To incorporate this comment into the revision, we have included the STM images to the supplementary information and added the following sentences into the main text:

“The observed phase segregation process relies on the inherent asymmetric adatom-substrate exchange barriers to result in the enrichment of Au atoms at the surface. We envision the broader applicability of this phase separation process because size differences between

constituent atoms in alloys can typically result in the different adatom-substrate exchange barriers between dissimilar atoms. However, stoichiometric, intermetallic compounds may have less tendency than solid solutions to undergo such a phase separation process because the strong interatomic bonding in intermetallic compounds can make the surface more stable at the elevated temperature, thereby reducing the number of active sources (e.g., atomic steps, kinks, ledges) to form adatoms via step-edge detachments. To further confirm this feature, we also performed scanning tunneling microscopy experiments by annealing intermetallic compound $\text{Cu}_3\text{Au}(100)$ at $\sim 600^\circ\text{C}$ in ultrahigh vacuum (Supplementary Note 11). As shown in Supplementary Fig. 12, the STM images indicate that the overall surface density of clusters is lower than that by annealing the $\text{Cu-10at.\%Au}(100)$ thin film (as shown in Fig. 1(a)) despite the higher Au content in the intermetallic Cu_3Au crystal.” (please see page 12, lines 19-31)

Supplementary Fig. 12: STM images obtained from the $\text{Cu}_3\text{Au}(100)$ annealed at $\sim 600^\circ\text{C}$ in ultrahigh vacuum. (a) A typical surface area that is nearly free of clusters, (b) a separate surface area showing the presence of clusters.

2. Can the absence of a beam effect be quantified beyond "likely" (line 228)?

Reply: This is a good point. The procedures for estimating the beam effects were well-established by Egerton [1], and have been used to estimate the potential effects on the Au/Cu related systems [2], the quantitative analysis suggests that the electron beam has a negligible effect on the dynamics of Au particles. We have followed the established procedures to quantify the possible beam effects on this Cu-Au system, and the analysis leads to the same conclusion as that in [2], i.e., the electron beam has negligible effects on the observed phenomenon.

According to Egerton et al., possible electron beam effects on TEM observations include charging, heating, atom displacement, sputtering, and radiolysis, below we analyze the effects on the Cu-Au systems one by one.

1) Charging: Charging effects are expected to be trivial due to the high conductivity of both Cu and Au.

2) Heating: One of the major effects that contribute to the heating of the system is the inelastic collision of the beam electrons and Cu/Au electrons, which is estimated by, $\Delta T = I \cdot \langle E(eV) \rangle 4\pi\kappa\lambda [0.58 + 2\ln(2R_0/d)]$ (1). in which I is the electron beam current, $\langle E(eV) \rangle$ is the average energy loss per inelastic collision, λ is the mean free path for in elastic scattering, κ is the thermal conductivity of the material, d is the incident beam diameter, R_0 is the heat conduction distance (1). In our experiments, the beam current is ~ 2.1 nA, $\langle E(eV) \rangle$ is 50~100 eV (plasma peak of the electron energy loss spectrum), ~ 314 W/m/K (for Au), ~ 385 W/m/K (for Cu substrate), λ is ~ 84 nm (for Au), ~ 100 nm (for Cu substrate), beam diameter d is ~ 600 nm, heat dissipation radius R_0 is ~ 3 mm. Putting the values of these parameters into the equation gives a temperature rise of < 1 K, which is negligible relative to the annealing temperature (600°C).

3) Atomic displacement: The atomic displacement is typically considered to be related to elastic knock-on of the electron beam, generating vacancies and interstitial atoms. The maximum energy that can be adsorbed by the system can be estimated by (1), $E_{\max} = E_0(1.02 + E_0/106)/(465.7A)$, where E_0 is the incident energy of electron in eV, A is the mass number of the element (1). In our experiments, E_0 is ~ 300 keV, A is 197 for Au. This gives a maximum transfer energy of ~ 4.3 eV which is well below the threshold displacement energy E_d of Au ~ 34 eV (1). Therefore, displacement effect is also trivial in this system.

4) Sputtering: The sputtering can be triggered when the E_{\max} is larger than the threshold value of sample (E_s) which can be estimated from sublimation energy (1). Considering E_{\max} (~ 4.3 eV) $>$ E_s of Au (~ 3.8 eV/atom), the sputtering is a possible event in our experiments, the rate of which can be estimated by (1), $S = (J/e)(Z^2/AE_0)(1/E_s - 1/E_{\max})(3.54 \times 10^{-17} \text{ cm}^2)$, where J/e is the electron dose rate, Z is the atomic number (79 for Au). Putting the dose rate of $\sim 3 \times 10^{19}$ electrons/cm²/s (~ 4 A/cm²) into the equation gives a sputtering rate of ~ 0.0032 monolayer/s. Given the typical TEM image acquisition rate (2 frames per second) in our experiments, the sputtering effects on the TEM observations are negligible.

In the revision, we have incorporated the above analysis on electron beam effects into supplementary information (please see Supplementary Note 9), and added the following sentences into the main text:

“The possible electron beam effects including charging, heating, atom displacement, sputtering, and radiolysis, are concluded to be negligible in our observations (Supplementary Note 8), consistent with previous work (Ref 2)”

References:

1. Egerton, R.; Li, P.; Malac, M., Radiation damage in the TEM and SEM. *Micron* 2004, 35 (6), 399-409.
2. He, Y.; Liu, J.-C.; Luo, L.; Wang, Y.-G.; Zhu, J.; Du, Y.; Li, J.; Mao, S. X.; Wang, C., Size-dependent dynamic structures of supported gold nanoparticles in CO oxidation reaction condition. *Proceedings of the National Academy of Sciences* 2018, 115 (30), 7700-7705.

3. Is it possible that hydrogen plays some role?

Reply: This is a great question that was also echoed by reviewer 2. The purpose of using hydrogen is to provide a reducing environment to remove any native oxide and maintain the surface cleanliness. This is confirmed by EELS measurements showing the absence of O signal from the Cu-Au film annealed in H₂. To further confirm whether the H₂ gas flow has any effect on the observed surface segregation, we recently employed an ambient-pressure X-ray photoelectron spectroscopy (AP-XPS) system to monitor the surface composition evolution of

Cu₃Au(100) during ultrahigh vacuum (UHV) annealing and annealing in 0.1 Torr of H₂ gas flow. As shown below (and also see Supplementary Note 10 and Supplementary Fig. 12), no noticeable differences in the surface composition can be observed between the UHV and H₂ gas flow [1]. This is consistent with other studies showing the high dissociation barriers of H₂ molecules on both Cu and Au surfaces [2-4]. Even for atomic hydrogen, it bonds weakly to Cu and Au, and desorbs from Au surfaces at the temperature of above ~ -163°C [5] and from Cu surfaces at the temperature of ~88°C [6]. Our in-situ TEM experiments were performed at ~ 600°C, much higher than the desorption temperatures for atomic hydrogen on Cu and Au surfaces.

Supplementary Figure 11. Comparison in surface Au composition of Cu₃Au(100) for the annealing at 600°C between ultrahigh vacuum (UHV) (a) and 0.1 Torr of H₂ gas flow (b). Spectra were taken with the photon energy of 400 eV. The Au 4f region consists of two contributions corresponding to Au-4f_{7/2} and Au-4f_{5/2}, respectively. Both contributions can be deconvoluted into two components, i.e., pure Au (Au_p) and alloyed Au (Au_a). The annealing under the H₂ gas flow does not induce any notable differences in pure Au concentration, as indicated by the Au_a/Au_p ratio of integrated peak areas of each Au species.

In the revision, we have included the AP-XPS results into the supplementary material (Supplementary Note 10 and Supplementary Fig. 11). We have clarified this point in the main text as follows:

“In addition, the H₂ gas flow has negligible influence on the observed phase segregation other than providing a reducing environment to maintain the surface cleanliness. This is confirmed by EELS measurements showing the absence of oxygen signal from the Cu-Au film annealed in H₂. To further confirm whether the H₂ gas has any effect on the surface segregation, we also employed ambient-pressure X-ray photoelectron spectroscopy (AP-XPS) to monitor the surface composition evolution of Cu₃Au(100) during annealing at ~ 600°C under ultrahigh vacuum (UHV) and 0.1 Torr of H₂ gas flow [1]. No noticeable differences in the surface composition can be observed between the UHV and H₂ gas flow (Supplementary Note 10 and Supplementary Fig. 11). This is consistent with other studies showing the high dissociation barriers of H₂ molecules on both Cu and Au surfaces [2-4]. Even for atomic hydrogen, it bonds weakly to Cu and Au, and desorbs from Au surfaces at the temperature of ~ -163°C [5] and from Cu surfaces at the temperature of ~88°C [6], both of which are much lower than the annealing temperature in our in-situ experiments.” (please see page 9, lines 24-31 and page 10, lines 1-9)

References:

1. C. R. Li, Q.Q. Liu, J. A. Boscoboinik, and G. W. Zhou, Tuning the surface composition of Cu₃Au binary alloy, *Phys. Chem. Chem. Phys.* 2020,22, 3379-3389
2. B. Hammer and J. K. Norskov, Why gold is the noblest of all the metals, *Nature*, 1995, 376, 238–240.
3. L. Stobiński and R. Duś, Model of atomic hydrogen adsorption on thin gold film surface, *Vacuum*, 1994, 45, 299–301.
4. J. Greeley and M. Mavrikakis, Surface and Subsurface Hydrogen: Adsorption Properties on Transition Metals and Near-Surface Alloys, *J. Phys. Chem. B*, 2005, 109, 3460–3471.
5. M. Pan, A. J. Brush, Z. D. Pozun, H. C. Ham, W.-Y. Yu, G. Henkelman, G. S. Hwang and C. B. Mullins, Model studies of heterogeneous catalytic hydrogenation reactions with gold, *Chem. Soc. Rev.*, 2013, 42, 5002–5013.
6. I. Yasumori, N. Momma and M. Kiyomiya, Mechanism of Hydrogen Adsorption and Hydrogen-Deuterium Equilibration on Copper Surface, *Jpn. J. Appl. Phys.*, 1974, 13, 485.

4. How low is the oxygen content of the film (referring to bulk, not surface oxide)? Such oxygen could react with incoming hydrogen.

Reply: We agree with this comment. The alloy films were prepared using an ultrahigh vacuum e-beam system, and the oxygen content in the as-prepared films should be negligible. The transfer of the thin film sample to the TEM can induce some oxygen (and native oxide) to the film, which can be removed during annealing by reacting with incoming hydrogen. We have performed in-situ electron energy loss spectroscopy (EELS) analyses to probe the oxygen content in the bulk of the thin film.

Shown below are representative EELS O-K and Cu-L spectra obtained in-situ. The unannealed film indeed contains a small amount of oxygen, as shown by the measured intensity of the O K edge. The intensity of the O K peak disappears in the well-annealed film. The measured Cu L spectra do not show much difference before and after annealing, suggesting the negligible amount of native oxide. The EELS results are also consistent with our recent ambient-pressure X-ray photoelectron spectroscopy (AP-XPS) measurements performed on a bulk crystal of Cu(110), which showed that bulk-dissolved oxygen in Cu can be completely deoxygenated by the flow of H₂ gas at ~580°C to form H₂O molecules that spontaneously desorb from the surface, resulting in an atomically clean Cu surface at the elevated temperature [1].

Reference:

1. C.R. Li, P.H. Zhang, J.Y. Wang, J.A. Boscoboinik, G.W. Zhou, "Tuning the Deoxygenation of Bulk-Dissolved Oxygen in Copper", *Journal of Physical Chemistry C* 122, 8254-8261(2018)

Supplementary Fig. 3(c, d): EELS O-K (a) and Cu-L_{2,3} (b) edges obtained *in situ* from the Cu-Au(100) thin film before and after annealing at $T \sim 600^\circ\text{C}$. The black ones correspond to the film before annealing, and the red ones are obtained from the sample after the annealing for ~ 30 min.

We have clarified this point by incorporating the EELS results into the supplementary information. In addition, the following sentences are added into the main text at two places to clarify this point:

"The Cu-Au films are then annealed at 350°C and 1×10^{-3} Torr of H_2 gas flow to remove native oxides and generate faceted holes (Supplementary Fig. 3(a)), where the complete removal of native oxide is confirmed by electron diffraction (Supplementary Fig. 3(b)). EELS experiments are also performed to ensure that the thin films annealed in H_2 gas flow are free of oxygen (Supplementary Figs. 3(c, d))." (please see page 3, lines 8-12)

"In addition, the H_2 gas flow has negligible influence on the observed phase segregation other than providing a reducing environment to maintain the surface cleanliness. This is confirmed by electron diffraction and EELS analyses showing the absence of oxygen in the Cu-Au film annealed in H_2 (Supplementary Fig. 3). This is also consistent with our ambient-pressure X-ray photoelectron spectroscopy (AP-XPS) measurements, which showed that any bulk-dissolved oxygen in Cu can be completely deoxygenated by the flow of H_2 gas at $\sim 580^\circ\text{C}$ to form H_2O molecules that spontaneously desorb from the surface, resulting in an atomically clean Cu surface at the elevated temperature⁴⁶." (please see page 9, lines 21-28)

5. Although the exposure temperature is well above any order-disorder transition, is it

possible that the type of pairwise atomic interaction implied by that ordering tendency plays some role in the phenomenon?

Reply: This is a great point. As described earlier in our response to comment 1, the ordered phases of Cu-Au intermetallic compounds are stable to temperatures of $\sim 390^\circ\text{C}$. Our in-situ TEM observations also confirmed that an ordered Cu_3Au -like surface alloy develops by annealing the Cu-10at.%Au film at $\sim 350^\circ\text{C}$ (please see our response to comment 1 and Supplementary Fig. 3). The ordered surface alloy formation is induced by the interplay between the chemical ordering tendency to form Cu-Au bonds and surface segregation of Au atoms, where the latter favors the occupation of neighboring lattice sites by the same atomic species at the surface sites while ordering causes exactly the opposite. The ordered Cu-Au alloy has improved surface stability and less tendency to undergo the phase separation because the pairwise atomic interaction results in the favored Cu-Au configuration. By contrast, the observed phase separation at 600°C suggests that the pairwise Cu-Au atomic interaction is significantly weakened at the temperature above the order-disorder transition, for which the disparity in the adatom-substrate exchange barriers results in the enrichment of Au adatoms at the surface that subsequently aggregate into Au clusters. The comparative observations made by annealing the Cu-Au films at the temperatures below and above the order-disorder transition temperature indicate that the phase separation can be suppressed by Cu-Au pairwise interactions.

We have incorporated this comment into the revision as follows:

“The ordered phases of Cu-Au intermetallic compounds are stable to temperatures of $\sim 390^\circ\text{C}$. Our in-situ TEM observations also confirmed that an ordered Cu_3Au -like surface alloy develops by annealing the Cu-10at.%Au film at $\sim 350^\circ\text{C}$ (Supplementary Fig. 3). The ordered surface alloy formation is induced by the interplay between the chemical ordering to form Cu-Au bonds and the tendency for surface segregation of Au atoms, where the latter favors the occupation of neighboring lattice sites by the same atomic species at the surface sites while ordering causes exactly the opposite. The ordered Cu-Au alloy has improved surface stability and less tendency to undergo the phase separation because the pairwise atomic interaction results in the favored Cu-Au configuration. By contrast, the observed phase separation at 600°C suggests that the pairwise Cu-Au atomic interaction is significantly weakened above the order-disorder transition temperature. The comparative observations made by annealing the Cu-Au films at the temperatures below and above the order-disorder transition temperature indicate that the phase separation can be suppressed by Cu-Au pairwise interactions at the lower annealing temperature” (please see page 12, lines 5-18)

6. I am not really in agreement with calling this "dealloying", as that is bound to cause confusion, and the physics of this are quite different from what is normally understood by dealloying.

Reply: Thank you for this comment, we have replaced “dealloying” with “phase separation”.

REVIEWERS' COMMENTS:

Reviewer #1 (Remarks to the Author):

After addressing the raised issues, the revised manuscript has exhibited an improved quality now. Thus, I recommend the acceptance of this revised manuscript in Nat Commun.

Reviewer #2 (Remarks to the Author):

The authors have answered all open questions.

I recommend this manuscript for publication.